# Reconstructions of Droughts in Germany since 1500 - combining hermeneutic information and instrumental records in historical and modern perspectives

Rüdiger Glaser[1], Michael Kahle[1]

[1]Physical Geography, Institute for Socio-Environmental Studies and Geography, University of Freiburg, 79098, Germany

*Correspondence to*: Rüdiger Glaser (ruediger.glaser@geographie.uni-freiburg.de)

**Abstract**. The present article deals with the reconstruction of drought time series in Germany since 1500. The reconstructions are based on historical records from the virtual research environment
tambora.org and official instrumental records. The historical records and recent data were related with each other via modern indices calculations, drought categories and their historical equivalents.

Historical and modern written documents are also taken into account to analyse the climatic effects and consequences on environment and society. These pathways of effects are derived and combined with different drought categories.

The derived Historical Precipitation Index (HPI) is correlated with the Standardized Precipitation Index (SPI). Finally, a Historical Drought Index (HDI) and a Historical Wet Index (HWI) are derived from the basic monthly hygric indices (PI) since 1500. Both are combined for the Historical Humidity index (HHI). On this basis, the long-term development of dryness and drought in Germany since 1500, as well as mid-term deviations of drier and wetter periods and individual extreme events are
presented and discussed.

**Keywords**: Historical Drought Index, Historical Humidity Index, Pathways of Effects, Extreme Droughts

## 1 Introduction

In Central Europe exceptionally extreme droughts such as in 2018, 2015 and 2003 have occurred quite often in recent years (Erfurt et al. 2019, Blauhut et al. 2015, 2016). The comparatively dense sequence raises the question to what extent this is another indicator of climate change. The overall damages add up to billions, with agriculture and forestry being primarily affected. Ongoing droughts also have negative consequences on water balance and water supply, ecology, economy and society.
Over the last years, health issues as well as the impacts on infrastructure and transportation have been discussed (Bachmair et al., 2016, Stagge et al. 2015, Van Dijk et al., 2013).

In addition to many climatological, ecological and social specifications, long-term reconstructions of droughts are helpful for a better, more holistic understanding. They contribute significantly to answering questions about long-term trends, accumulations, recurrence times and the variability of
extreme events. There is also evidence about societal contextualization, especially the impacts and responses on environment and societies, which have changed fundamentally through time (Erfurt et al. 2019).

Droughts are generally referred to as periods of extremely dry weather that persist long enough to cause a severe deficit in the water balance, which in turn causes environmental and social impacts and damages (Wilhite 2000, Glaser & Erfurt 2019). From a statistical point of view, a drought is an exceptional event with a rare recurrence probability (Benestad 2003). According to a widely used scheme, droughts are subdivided into four types that reflect their chronological development: A meteorological drought describes a period of considerable precipitation deficit, usually in comparison with a reference period. High air temperatures and wind speeds, intensive solar radiation and cloudless skies can aggravate the precipitation deficit (Wilhite 2000). With continued duration, the amount of plant-available soil water is reduced, with negative effects on plant growth and harvest yields. In this case, we speak of an agricultural drought (Bernhofer et al., 2015). If the drought continues to progress, reduced surface runoff, sinking water levels and ultimately sinking groundwater levels occur - a so-called hydrological drought. Lastly, extremely low groundwater levels or baseline flows are referred to as groundwater droughts. Additionally, the term socio-economic drought is used when it comes to impacts on people and the environment. The degree of severity depends on the  assets, adjustment options and resilience of the affected society (Wilhite 2000, McKee et al 1993).

Droughts can be defined numerically according to very different criteria, which result in a large number of indices. They differ in the type of input data, temporal and spatial coverage and the consequences for different sectors. While the input data used for meteorological droughts are temperature and precipitation, assessments of hydrological droughts are based on gauging data, groundwater levels and runoff. The timeframes range from days over weeks and months to years (Bernhofer et al., 2015). Similarly, the size of the study area varies according to the question.

Common drought indices include the Standardized Precipitation Index (SPI), the Standardized Precipitation Evapotranspiration Index (SPEI) and the Palmer Drought Severity Index (PDSI). In the US, the PDSI is the most common drought index, which is also used as the basis for the US Drought Monitor (Palmer 1965, McKee et al 1993, Vicente-Serrano et al 2010, Zargar et al 2011, Svoboda et al 2012). The German Drought Monitor (Dürremonitor Deutschland) represents the current monthly status of the soil in five drought categories (Zink et al 2016, DWD 2019). Moreover, drought assessments also examine ecological and social consequences (Stahl et al., 2016).

The strengths and weaknesses of the SPI are widely discussed in literature (Briffa et al 1994, Cook et al 2015). Cook et al. (2015) or Mikšovský (2019) for example used the more complex scPDSI, which needs inter alia soil water information and additional temperature data. The present study is based on the SPI because of its wide distribution, simple calculation and its ability to integrate historical events.

Numerous studies on droughts with an explicitly historical perspective have been presented in recent years, for example by Nash & Grab (2010), Gil-Guirado et al (2016), Brazdil et al. (2018, 2019) and Erfurt et al. (2019), implementing a wide range of content and methodological aspects. The spectrum ranges from analyses of outstanding individual years (Wetter et al., 2014) and the derivation of regional time series in different climatic zones (Noone et al 2017, Kiss 2017, Dobrovolný et al., 2018, Nash et al., 2019) to a focus on drought effects (Glaser et al., 2017).

The aim of the present contribution is the reconstruction of long-term drought time series in Germany since 1500 based on written records. For this purpose, a rating scheme was developed in order to correlate recent parameters and criteria of drought valuations with historical ones. For this purpose, various drought indices and drought categories were derived and evaluated. The long-term development was assessed, particularly the question to what extent the current developments differ from the previous phases.

## 2 Data

The analysis is based on two main comprehensive data sets. The first data set - available in the virtual research environment *tambora.org* (tambora.org, Riemann et al. 2016, Glaser et al. 2018) - consists of written documents related to weather and climate as well as the impacts and consequences on the environment and society of Central Europe. The second, modern data set comprises official precipitation data for Germany from 1881 onward, provided by the German Weather Service (DWD).

The approximately 330,000 historical records in *tambora.org* for Central Europe are taken from weather diaries, chronicles, pamphlets, official reports and newspapers. Other media such as flood marks, hunger stones, pictures and lyrics supplement these. The 330,000 coded records in *tambora.org* are represented as blue dots in Fig. 1, while the red dots represent the 54,000 records indicating precipitation and specific information regarding dryness and droughts. Additionally, the green dots indicate the 12,600 records describing the impacts and consequences of dryness, drought and lack of precipitation. Such descriptions include water shortages, low water levels of larger rivers, fish kills, forest fires, emergency slaughteries, crop failures or prayers for rain. In total, the information covers large parts of Central Europe. The southwest and the center as well as the eastern parts of Germany are particularly well depicted, but also the larger river systems such as Main, Rhine and Elbe. Additionally, the spatial distribution concentrates around the cultural, political, economic and religious centers such as Nuremberg, Cologne, Leipzig, Erfurt, Hamburg and Mainz as well as other larger cities and monasteries. The coastline is also well represented, specifically the harbour locations like Hamburg, Lübeck and Rostock. Temporal coverage is very good, with information for every month since 1500. As expected, average and inconspicuous months are less documented than more extreme ones.

All records in tambora.org are numerically coded, comprising spatial, temporal and content aspects. In addition to the coded events, the original text quotes are also included in the database so that the overall context and the coding can be traced for each record.

[Fig. 1: Spatial distribution of all records referring to Central Europa in *tambora.org* since 1500 (blue), precipitation, dryness and drought records (red), and impacts and consequences (green).]

The second, modern data set used for the analysis consists of the official precipitation data for Germany from 1881 onward. These values, monthly precipitation measurements in mm, recorded, averaged and provided by the DWD (Deutscher Wetter Dienst, the official German Weather Service) from the national official network stations, representing the area of modern-day Germany. This study also draws upon the official drought categories D0-D4, their classification and their definitions based on SPIs by the DWD (2019). The relation between D0-D4 and the relevant SPIs are indicated in Table 1. The relation is based on duration and intensity defined via different thresholds.

## 3 Methods

The conceptual design of the analysis is given in Fig.2. It illustrates the single steps and the workflow as a whole. Each individual step is described in the following subchapters in detail.

125

[Fig. 2: Conceptual design of the analysis, illustrating the workflow and the single steps deriving the specific indices and their relations]

Tab.1 Abbreviations of the developed indices

130

| Abbreviation | Description | Range (dry .. wet) | Remark |
|---|---|---|---|
| PI | Precipitation Index | -3 .. +3 | |
| SPI | Standard Precipitation Index | -x.x .. +x.x | related to normal distribution, theoretically all values possible, usually less than 4 |
| HSPI | Historical Standard Precipitation Index | -x.x .. +x.x. | depends on HPI and calibration of slopes |
| HPI | Historical Precipitation Index | -15 .. +15 | theoretically -36 .. + 36 |
| MDI | Modern Drought Index | 4 .. 0 | according to DWD drought categories |
| HDI | Historical Drought Index | -4.0 .. 0.0 | |
| HWI | Historical Wet Index | 0.0 .. 4.0 | |
| HHI | Historical Humidity Index | -4.0 ... +4.0 | Combines HDI & HWI |

**3.1 Derivation of the monthly Precipitation Index (PI) since 1500**

The hygric indices (PI) were derived from the written evidence of the tambora sources via semantic
135    profiles, a method well established in historical climatology (Glaser 1991, 1996, 2013, Glaser & Riemann 2009, Pfister 1999, Brazdil et al., 2005). Therefore, direct hygric indications as well as the descriptions of impacts and consequences are hierarchically ordered according to their intensity and assigned to the appropriate index value. A seven-scale index scheme, ranging from -3 to +3 with index 0 representing the average situation, has proven to be appropriate for the classification of
140    historical records (Glaser 1991, 1996, Glaser & Riemann 2009, Glaser 2013).

The hierarchical class assignment and its typical indicators for the negative precipitation indices (PI) -1 to -3 are presented as follows:

Index -1 is indicated by descriptions of a beginning rainfall deficit. There are often indications of higher damages relating to the harvest of rain-sensitive products such as hay, vegetables and other garden products.

Index -2 relates to a longer duration of lack of precipitation, prolonged heat and dryness. Average crop losses for main crops are reported as well as low water levels in smaller bodies of water and reduced spring fills. Heat stress on plants, premature leaf discoloration and the death of plant parts are observed, also dry cracks in soils, occasional forest fires and the impairment of infrastructure, for example related to shipping and water mills.

Index -3 represents extreme dryness revealing a chain of effects: After a prolonged period of dryness and heat, the agrarian consequences include severe crop losses and even harvest failures as well as emergency slaughteries due to fodder shortages. If the dryness lasts for weeks, several months or even seasons, there are integrating effects like low water levels in greater lakes, ponds and larger river systems as well as the drying up of springs and wells. In addition, reports of excessive water shortage and the appearance of "hunger stones" are common. Ecological impacts include visible heat stress of the vegetation, premature leaf discoloration and the withering of plants; dry cracks in soils, dust veils, effects of wind erosion are indicated. There are diverse descriptions of a shift of the phaenological phases, e.g. early flowering, ripening and harvest, but also expressions like "wine of the century" reports of forest fires and fish kills. The impairment of infrastructure, especially the termination of shipping and the failure of mills are frequently mentioned. The direct consequences for human health are also documented, e. g. through indications of heat stress, increased death rates, the outbreak of epidemics and hunger crisis due to a lack of food. In addition, the reports include price increases and speculations.

Authorities´ reactions range from restrictions and regulations on water access to the declaration of a state of emergency. Societal reactions like supplications, processions, pilgrimages, increasing irrational explanations and interpretations are quite common. The sources also report begging, moving around in order to seek food and protests, theft, looting, robbery and social excesses. These integrating effects allow conclusions to the preceding months, and in many cases, the exact dates of meteorological droughts are indicated by the name day of saints.

The indexing process is similar to modern classifications and definitions of drought categories. Such modern drought categories also take into account the descriptions of impacts and societal consequences and reactions (McKee 1993, NDMC 2018).

Weather diaries with daily records exist for more than 60 years from the period 1500 to 1800, containing precipitation days (Lenke 1960, Klemm 1964, 1967, Glaser & Gudd 1996, Glaser 1996, Glaser 2013). These records are compared with modern precipitation data on a monthly scale, enabling a comparison of numerical rainfall data with the classified written evidence, which serves as an additional verification and validation of the index levels (PIs).

Tab.2: Selected Observers, Location and Periods with Daily Weather Entries 1500-1800

| Observer | Location | Period | Percentage of daily data |
|---|---|---|---|

| | | | |
|---|---|---|---|
| J. Stoeffler, | Tübingen | 1507-1530 | 80% |
| Johannes Indagines | Rheingau | 1517-1519 | 90% |
| NN | Dresden | 1580/1582 | 100% |
| Leonhard III Treuttwein | Fürstenfeld | 1587-1593 | 85% |
| Kilian Leib | Rebdorf | 1513-1531 | 85% |
| Hermann IV | Hessen-Kassel | 1621-1650 | 100% |
| Gottfrid Wilhelm Leibniz | Hannover | 1678 | 100% |
| Friedrich Hoffmann | Halle | 1700 | 100% |
| Camerarius | Tübingen | 1712-1715 | 85% |

The positive hygric index corresponds to the humid and wet situations and is derived in the same manner. The summary of the monthly PI for Germany from 1500 onward is given in Fig. 3. The monthly PI reveals a differentiated picture of drier and wetter periods since 1500. The data is available via Glaser & Kahle (2019).

[Fig. 3: Summary of the monthly precipitation index (PI) for Germany from AD 1500-2018]

### 3.2 Derivation of historical pathways and drought categories and their mapping to modern definitions

The consequences and effects of drought on environment and society recorded in historical sources - here referred to as pathways - resemble the structure and classification schemes used in modern drought classifications (Bernhofer et al. 2015, U.S. Drought Monitor, DWD 2019): A precipitation deficit is followed by a specific pathway. It first appears on the agricultural, then the hydrological and finally the socio-economic level - a development reflected by modern drought definitions (Nash et al. 2019, Erfurt et al. 2019).

These time- and intensity-related chains of effects can be derived from the historical sources as characteristic pathways: First of all, the absence of rain and first signs of dryness and a beginning drought are usually described very precisely in historical sources. Often these are provided with time information, especially duration, beginning and end of drought effects. Very often information is given on the phenological phases, particularly the prematurity of flowering, but also field cropping and harvest dates (Freiburger Zeitung 1834). With increasing drought, the consequences for agriculture like crop damages and crop failures, especially in rain-sensitive horticultural products and

hay, are described. As the drought progresses, both the number of descriptions and their differentiation increase, including emergency slaughters for lack of food and the use of emergency reserves. At this stage, descriptions of the water balance also appear frequently: low water levels in water bodies, subsidence of spring discharges, drying up of small wells. Effects on the environment

such as forest fires, fish dying, algae blooms and various forms of soil degradation such as deflation and dry cracks complete the picture (Brooks & Glasspole 1922, DWD 1947a, b, Dürr 1986). The explanations are now also supplemented by indications of infrastructural problems, particularly regarding low water levels and the operation of mills. Health consequences are also recorded, including an increased mortality following the outbreak of epidemics, often due to poor water

quality. Harvest losses lead to price increases and subsequently to hunger (Nees & Kehrer 2002). Religious rites such as prayer services for rain or processions, but also official measures such as water rationing are taken. If the drought persists, the conditions described become more acute. In the historical context, especially after famine crises and epidemics, there are social excesses such as looting, robbery, persecution of minorities and excluded groups as well as migration movements

(Glaser et al., 2017, 2018). Fortunately, these are lacking in the modern context in Europe after 1950. However, extreme droughts in the post-war period 1947 and 1949 were also accompanied by protests and strikes as a result of the special circumstances (DWD 1947a, Erfurt et al., 2019, Brazdil et al., 2016).

These chains of effects reflecting the duration of a drought period are understood as pathways. Their

grades were also classified as drought categories (see last column of Tab. 3). As these are very similar to descriptions of the consequences and implications of modern classifications, it is possible to parallelize them on a hermeneutical basis.

### 3.3 Determination of modern SPI and mapping of recent drought categories

The modern Standardized Precipitation Index (SPI) was calculated from the official precipitation values for Germany 1881 - 2018 provided by the DWD, Climate Data Center (2019), using the package 'SCI' (Gudmundsson & Stagge 2016, Stagge et al. 2015, 2016). The different SPIs were calculated for the corresponding time periods of one to twelve months as SPI1 to SPI12.

The drought categories D0-D4 and the characterization of droughts as well as the duration and the

description of the consequences were also taken over from the scheme of the DWD (DWD 2018), see Tab.1, first to fourth column.

### 3.4 Derivation of the Historical Precipitation Index (HPI)

The derivation of the Historical Precipitation Index (HPI) is based on the monthly Precipitation Indices (PI). We calculated the HPI as the sum of the PIs of the corresponding number of the relevant

months. This was done for time windows from one to twelve months in order to map the accumulative effects of dryness and lack of rainfall, analogous to the SPI. For example, HPI3 of June results from the sum of the PIs April to June. We also included positive values for humid and wet conditions.

### 3.5 Correlation of the Historical Precipitation Index (HPI) with the modern SPI

To compare the HPI with the modern SPI, a correlation analysis for the overlapping period 1881-1996 was applied. The results show a very high correlation of 0.65 to 0.74 between SPI and HPI. The two parameters (HPI vs SPI) are highly correlated. The strength of the statistical relationship and its shape are shown in Fig. 4.  There is also an relation between the specific slope and the duration. Therefore, we introduced a duration dependent scale factor.

[Fig. 4: Strength and shape of the relationship between SPI and HPI 1881-1996 for the duration of one to twelve months]

To integrate the duration effect, this derived scale factor was used. The comparison of the accumulated values allows the identification of a factor dependent on the number of months, which we used to adjust the scaling (Fig. 5).

[Fig. 5: Duration and scale factors of the relationship between SPI and HPI for the duration of one to twelve months]

We solved the equation and applied the following inverse function to map the class boundaries given in SPI values (see Tab.1, column 1) to HPI values (Tab.1, column 5).

$$HPI_i = SPI_i \cdot i^{\frac{1}{\sqrt{3}}}$$

### 3.6 Derivation of HSPI time series from 1500

In a further step the scale factors were applied to the Historical Precipitation Index (HPI) since 1500 to derive the Historical SPIs (HSPIs). Examples of the transformed HSPIs are shown as HSPI3, HSPI6 and HSPI12 (Fig. 6).

[Fig. 6: HSPI3, HSPI6 and HSPI12 for Germany since 1500 for the duration of three, six and twelve months]


### 3.7 Derivation of the Historical Drought Index (HDI) and comparison with the Modern Drought Index (MDI)

For the determination of the Historical Drought Index (HDI), we stepwise interpolated the classes linearly defined in Tab. 1, using the negative SPIs with different durations (Fig. 7). These were
compared with the Modern Drought Index (MDI) of the DWD for the calibration phase 1881-1996. The correlation of $r^2 = 0.48$ underlines the strength of the relationship between the two variables.

[Fig. 7: Comparison of the Historical Drought Index (HDI) and the Modern Drought Index (MDI) for Germany 1881-1991]

We also used this approach to calculate the last 500 years.


### 3.8 Synopsis of SPI, HPI and numerical as well as hermeneutic drought categories

To synthesize and compare the different numerical indices, drought categories and hermeneutic classifications, we compiled the modern SPI, the historical and modern drought categories (HDI & MDI), duration classes, recent descriptions and the HPI as well as the historical description of consequences and impacts in Tab 3.

The starting point are the parameters and criteria used in the Drought Severity Classification (NDCM) 2018) or the Drought Index of the German Weather Service (DWD 2018). These include drought indices derived from measured data such as the SPI as well as the assessment of the severity of droughts in the form of drought categories. These follow the criteria used in the general drought classifications of an agricultural, hydrological and socio-economic drought and include information on duration. In addition, we described the effects and consequences in short text blocks.

Tab. 3.: Synopsis of SPI and HPI and numerical and hermeneutic drought categories

| SPI Acc. DWD (2018) | Drought-categories (HDI & MDI) Acc. DWD (2018) | Duration months Acc. DWD (2018) | Recent descriptions of the effects and consequences as well as the duration Acc. DWD (2018) | HPI (Historical Precipitation Index) | Historical descriptions of effects and duration |
|---|---|---|---|---|---|
| -0.1 to -0.99 (SPI1, SPI2) | almost normal (slight dryness) D0 | 1-2 | short-term dryness | 0 to -1.5 (HPI1, HPI2) | low rainfall, heat and drought, possible first consequences for agriculture and yields |
| -1.0 to -1.49 (SPI1, SPI2) | moderate drought D1 | 1-2 | meteorological drought: one to two months drier than usual | -1.5 to -2.5 (HPI1, HPI2) | lower crop impact on main crops, failures in rain-sensitive horticultural products and hay, better wine quality |
| -1.5 to -1.99 (SPI2 - SPI4) | severe drought D2 | 2-4 | agricultural drought: two months and longer dry, crop losses | -2.5 to -4.5 (HPI2 - HPI4) | crop losses on main crops, emergency slaughter for lack of food, prematurity phenological phases, drying up springs, low water levels, mill arrest, forest fires, problems with water supply, heat deaths, measures of the authorities, price increases, hunger, religious rites |
| -2.0 to -2.99 (SPI4-SPI10) | extreme drought D3 | 4-10 | hydrological drought: from four months, groundwater and level affected | -4.5 to -12 (HPI4 - HPI10) | crop failures, emergency slaughterings, strong premature phenological phases forest fires, fish dying, algal blooms, soil erosion drying of springs and wells, low water levels of large rivers, hunger stones, heat deaths, epidemics, price increases and speculation, measures of the authorities, hunger, religious rites, increasing irrational explanations |
| -3.0 to -4.0 (SPI10-SPI12) | extra-ordinary drought D4 | > 10 | socio-economic drought: from one year, water shortage slows down producing economy | -12 to -36 (HPI10 - HPI12) | ...begging, moving about, searching for food, food substitution, robbery, plunder, murder, emigration and emigration, social excesses (Century-events) |

This table is of manifold uses for cross validation and comparisons of modern and historical Indices, as well as comparisons of hermeneutic descriptions with numerical indices, shortcutting calculations and conversions.

**3.9 Derivation of the Historical Wet Index (HWI) and Historical Humidity Index (HHI)**

To include not only dryness and drought aspects, humidity has also been considered in the analysis by including the positive hygric indices as a Historical Wet Index (HWI). Its derivation was analogous to the class boundaries of the drought categories. The dominating effects of the HDI and the HWI are combined into the Historical Humidity Index (HHI). The monthly results are shown in Fig. 8, along with the frequency-filtered signals for 1 and 5 years (Fig.8).

[Fig. 8: Historical Humidity Index (HHI) for Germany since 1500. The upper part represents the monthly HHI from January to December for each year and the overlaid filtered five-year low pass (black line). The lower part indicates the low-pass values represented as coloured schemes for five years and one year]

**3.10 Identification of extreme drought years since 1500**

        To identify and quantify the most outstanding extreme droughts over the centuries we compiled the different years according to the strength or intensity of the derived indices. All months with a HHI below the value -0.5 were selected and periods of consecutive dry months were clustered. For each of these periods, the minimum HHI value, its sum and its beginning, end and duration as well as the
HHI average were calculated. As usual in such compilations, the rankings of the top-century-events vary somewhat, according to the different indices and categories and their underlying calculations - weighting duration and intensities in different ways (see also Erfurt et al. 2019).

        In addition to the calculations, the chains of effects extracted from the written hermeneutic evidence were used to confirm extreme events. These are characterized by detailed evidence, emphasizing
strong impacts on agriculture, forestry, water circle and water supply as well as socio-economic effects like rising prices and hunger. Additionally, ecological effects were considered, e.g. wildfires, algae bloom, fish kills and soil erosion. There is also evidence about societal contextualisation, especially about the societies´ coping and adaptation strategies, which reveal their vulnerability and resilience capacity.

These aspects have changed through the ages: In the agrarian-feudal age, societies coped with drought very differently than during the industrialisation period when, for instance, migration became an option. There was also a great shift in the past 100 years: While the extremely vulnerable societies during and in the aftermath of the First and Second World Wars were severely affected by droughts, the consequences could be coped with during the modern episodes of 2003 and 2018
(Erfurt et al. 2019).

## 4 Results

In this paper we describe a conceptual and methodological approach for analysing drought events for Germany since 1500, based on written documents from the virtual research environment tambora.org and derived precipitation indices using hermeneutic approaches and pathway analysis. Different homogeneous index-based reconstructions of monthly and yearly drought time series for Germany since 1500 are presented connecting historical and modern times. This helps to interpret the long term development and to compare drought events across centuries.

The reconstructed monthly HHI f.e. clearly reveals the annual structures of dryness and wetness. It allows easy identification of dry months and longer periods of dryness and droughts. In total, the synopsis reveals the generally high variability of dryness and wetness through time. Additionally, the time series clearly show that not only summer, but also winter precipitation deficits occur. Springtime and autumns had also been affected. In this context, it is noticeable that in comparison to the analysis of summer droughts comparatively few studies on spring, autumn or winter droughts are available. The same can be stated for the humidity aspect.

To highlight the mid-term development, a 5-year frequency low-pass filter was applied. It emphasizes a somewhat higher variability in the first 150 years (1500 until 1650), including a very remarkable contrast between the dry period 1630-1635 and the moist phase 1646-1651, which was followed by a negative trend until 1700. This represents the most striking mid-term change of the last 500 years. The moist phase 1692-96 and the dry period 1740-1744 are also remarkable variations. Also towards the end of the 18th century very dry periods occurred during the 1770s and the 1790s. In addition, more dry months occurred in the period 1750 to 1911, as in the period after 1911. In the last few decades no significant trends have occurred, except for the most recent accumulation of extremes.

Even if the derived HHI shows some remarkable changes and a high variability on the 5-year scale, there is a remarkable stationarity of the annual humidity as expressed by the HSPI in the long-term 500-year perspective. In opposition to that there are remarkable shifts and seasonal trends on the longer scale, for instance winter humidity has increased while summer precipitation has decreased slightly during the last 150 years. There are also sections with comparable seasonal shifts and trends like an increase in winter humidity between 1590 and 1725 and in summer humidity between 1540 and 1690, as well a decrease in winter humidity between 1725 and 1800.

In order to further interpret the internal structure of this time series, a Fast Fourier Transformation was applied to the Historical Humidity Index (HHI) 1500-2018. This results in striking recurrence cycles at 22, 37 and 58 years. The same results can be found in the underlying indices, e.g. the HSPI.

As can be seen, outstanding drought events have appeared in all centuries since 1500. These are in general represented by a particularly large number and higher differentiation of sources. One example is the drought year 1540, where 41% of the 123 sources refer to agriculture, 17% to water, 11% to health, 10% to forest fires, 8% to soil and 8% to environmental and ecosystem issues. Other outstanding drought events are in the 16th century the droughts of 1503, 1522, 1567 and 1590. In the 17th century 1615, 1632, 1635, 1669 and 1681 had been described as exceptionally dry. The same can be stated for 1706 and 1719 during the 18th century. While in the 19th century, the years 1800, 1803, 1834, 1842, 1858, 1864 and 1893 are detected as drought years. For the 20th century this was the case in 1921, 1947, 1949, 1963 and 1976 and finally in the 21st century in 2003 and 2018 as can be seen in Table 4.

 Tab. 4: Table of outstanding drought periods since 1500 for Germany, in relation to different indices

| | Longer context of dryness | | Selected Indices | | | Reference based on tree rings (after Cook et al. 2015) |
|---|---|---|---|---|---|---|
| Year | Start | End | HHI min | HSPI.6 | HSPI.12 | Cook DE scPDSI |
| 1503 | Apr 1503 | Aug 1503 | -2.56 | -2.03 | -0.26 | -5.54 |
| 1522 | Apr 1521 | May 1522 | -4.00 | -3.69 | -3.81 | -0.17 (South-East: -1.4) |
| 1540 | May 1540 | Mar 1541 | -4.00 | -5.35 | -3.59 | -2.94 |
| 1567 | Apr 1566 | Jan 1568 | -2.99 | -2.36 | -2.48 | -2.13 |
| 1590 | Mar 1590 | Mar 1591 | -4.00 | -4.02 | -3.81 | -3.03 |
| 1615 | Jan 1615 | Mar 1617 | -3.81 | -3.69 | -3.81 | -1.91 (1616: -3.79) |
| 1632 | Mar 1630 | May 1633 | -3.31 | -2.69 | -3.15 | -1.71 |
| 1635 | Mar 1634 | Apr 1635 | -3.78 | -3.35 | -3.59 | -3.36 |
| 1669 | May 1669 | Feb 1670 | -3.15 | -3.69 | -3.15 | -3.30 |
| 1681 | Jul 1680 | Sep 1681 | -3.59 | -3.03 | -3.59 | -3.13 |
| 1706 | Jun 1705 | Jul 1707 | -4.00 | -4.35 | -4.47 | -2.07 |
| 1719 | May 1719 | Dec 1719 | -3.00 | -3.69 | -2.26 | -3.81 |
| 1800 | Nov 1799 | Jan 1801 | -3.98 | -3.69 | -3.58 | -3.06 |
| 1803 | Mar 1802 | Oct 1803 | -3.59 | -3.69 | -3.58 | -3.73 |
| 1814 | Dec 1813 | Jan 1816 | -3.54 | -3.36 | -3.37 | -0.51 (North: -2.4) |
| 1834 | Feb 1834 | Feb 1835 | -3.73 | -3.69 | -3.37 | -2.76 (1835: -4.44) |
| 1842 | Jan 1842 | Jan 1843 | -3.49 | -3.69 | -3.15 | -3.06 |
| 1858 | Feb 1857 | Oct 1858 | -3.00 | -2.69 | -2.92 | -4.64 |
| 1864 | Dec 1863 | Jan 1866 | -3.37 | -2.36 | -3.37 | -2.53 |
| 1893 | Mar 1893 | Mar 1894 | -4.00 | -4.69 | -3.37 | -4.17 |

| 1921 | Oct 1920 | Feb 1922 | -3.49 | -3.36 | -3.37 | -5.57 |
|------|----------|----------|-------|-------|-------|-------|
| 1947 | May 1947 | Oct 1947 | -2.46 | -2.36 | -2.04 | -3.96 |
| 1949 | Jun 1949 | Mar 1950 | -2.99 | -2.69 | -2.48 | -1.6 |
| 1963 | Jun 1962 | Jun 1963 | -2.92 | -2.69 | -2.70 | +0.37 |
| 1976 | Mar 1976 | Aug 1976 | -2.56 | -2.36 | -1.82 | -4.06 |
| 2003 | Mar 2003 | Dec 2003 | -2.47 | -2.33 | -1.76 | -1.36 |
| 2018 | Feb 2018 | Feb 2019 | -3.35 | -3.46 | -2.26 | - |

To visualize the outstanding single drought years, we also derived a „yearcloud"of classified years since 1500. It supports the comparison of the related extreme droughts to minor ones through time, especially between the modern and historical period.

[Fig. 9: "Yearcloud" of classified years since 1500. The average intensity is reflected by the color scheme and duration (month) by font size.]

## 5 Discussion and conclusion

The conceptual and methodological approach of this article using precipitation indices is well established in the research field of Historical Climatology (Glaser 1991, 1996, 2013, Nash & Grab 2010, Glaser & Riemann 2009, Pfister 1999, Brazdil et al., 2005, 2018). The analysis is based on written documents, which had been evaluated after a critical source analysis including hermeneutic principles (Glaser Glaser 1991, 1996, 2013).   Most of the written documents are part of the virtual research environment tambora.org. In the meantime there are a number of further comparable databases available, which also provide such data, likewise https://www.ncdc.noaa.gov/data-access/paleoclimatology-data/datasets/historical, Weather.org,  Pediflood (Barriendos et al., 2014), ICOADS (Woodruff et al., 2011), CLIWOC (García-Herrera et al. (2005),euroclimhist.ch (Pfister, 2015), EDII (Stahl et al. 2016) and REACHES 21 (Wand et al. 2018). As a standard in all of these data bases, all records are numerically coded, comprising spatial, temporal and content aspects like in tambora.org.

In a first step, a seven-scale index scheme had been applied to deduce a monthly Precipitation Index (PI). The derivation of such indices is also a fundamental step in Historical Climatology Research (Brazdil et al. 2005). In addition to monthly Precipitation Indices, many long-term paleoclimate reconstructions of precipitation, dryness and drought refer to indices like SPI, SPEI, PDSI or scPDSI (Bradzil et al. 2013, Dobrovolny et al. 2015, 2018, Erfurt et al. 2019). Some approaches are based on dendrological data series (Briffa et al. 2009, 1994, Büntgen et al. 2010, 2011, Cook et al. 2015) or follow a multi-proxy approach (Dobrovolny et al. 2018). Many of these elaborations relate to different regions and have a distinct temporal resolution. Some refer to the whole of Europe (Spinoni et al. 2015, Cook et al. 2015), many refer to the neighbouring Czech Lands (Bradzil et al. 2013, Dobrovolny et al. 2015, 2018), while Erfurt et al. (2019) have worked on South-West Germany, a sub-

area of Central Europe. Many paperrefer to the summer or the springtime situation or the vegetation period, some are based on yearly resolution. On the other side Cook et al. (2015) model soil conditions and humidity and temperature effects based on dendrological datasets. They present highly resoluted data, which sometimes looks from the aspects of the spatial dimensions of droughts to differentiated. In the Central European climatological context droughts can be regarded as large scale phenomena. Dobrovolny et al. (2018) follow a multi proxy approach, while Erfurt et al (2019) take early instrumental readings into account. Such differences in the methodological approach might explain differences in the resulting trends and developments through time as well as the rankings of the most outstanding events.

The long-term development and structures of the presented Historical Humidity Index (HHI) for Germany since 1500 are very similar to those of the precipitation reconstruction by Dobrovolny et al. (2015) and Mikšovský et al. (2019), which also was reconstructed for the neighbouring Czech lands based on documentary and instrumental data. There are some coincidences likewise the exceptional increase between 1630-1650 in the SPI time series, developed by Mikšovský et al. (2019), as well as the negative trend 1650-1700 with the dry period of 1692-96. Also the dry episodes 1770-1780 are well represented, as well as the long term trend 1850-1820. Other ones like the 1790s are less exposed. There are also correspondences with the moist periods in the 1690s and the 1940s. Most of the described periods also appeared in the precipitation reconstructions by Dobrovolny et al. (2015). There is also the striking 1630-1650 increase, also the decrease between 1650 and 1700, the very dry period around the 1860s and at the end of the 18th century, especially 1775-1785 as well as the long-term increase 1850-1920. While these go in parallel, other periods show an inverse development. Quite remarkable is the precipitation increase over the last decades.

With the presented approach it is possible to derive not only summer droughts but also droughts occurring in other seasons and longer periods, even multiyear droughts can be identified. It also allows the derivation of winter droughts, something which is less focused in literature. An exception is, for example, Pfister et al. (2006) with the analysis of hydrological winter droughts of the last 450 years in the Upper Rhine area.

Concerning the correlation of the derived long term trends in droughts with the temperature increase of the global warming trend, which is a major modern topic (Erfurt et al. 2019, Blauhut et al. 2016, Stahl et al. 2016), the results show, that there is no comparable outstanding development to the anthropogenic temperature trend over the last 200 years. This also corresponds to the findings of Sheffield et al (2012), Spinoni et al. (2015) and also of Noone et al. (2017) for Ireland. According to their analyses, drought frequency is increasing in southern Europe, with a reverse trend observed in northern Europe. Our study for Germany shows a remarkable long-term stationarity. The measured increase in drought events over the last decades must be regarded as normal from the long-term perspective. The historical variability is higher than the modern fluctuations since the 1950s. This is a remarkable statement for the long term evaluation of precipitation fluctuations especially in the context of the modern climate change debate and underlines the importance of long-term reconstructions.

In general, the method itself is transferable whenever the relevant regions allow the application of the SPIs. The results and conclusions presented in this article refer to the spatial outline of modern Germany, which shows homogeneity for this approach. In difference to modern drought-indices derived from instrumental records or multi-proxy approaches or dendrological datasets, the

445 approaches related to Historical Climatology directly take into account the descriptions of impacts and societal consequences and reactions. Such classification of hygric indications as well as the descriptions of impacts and consequences is here referred to as pathways.

The pathways are strongly coupled with the intensity and duration of the precipitation deficit. The agrarian, then the hydrological and finally the socio-economic consequences are presented in a 450 progressing chain of effects. Structurally, they correspond to today's drought definitions and classifications. The descriptions of impacts and consequences and the derived pathways of historical events are in most cases comparable to recent events (Erfurt et al 2019, Blauhut et al. 2016, Glaser et al. 2016, 2017, 2018, Brazdil et al. 2016, 2019 or Nash et al. 2019). Therefore, a parallelization is possible.

455 These statements apply to the pathways itself, but not to resilience and the underlying adaptation strategies. Obviously, these have developed over the centuries due to changes in social structures, assets and technical possibilities (Glaser et al. 2018, Camenisch & Rohr 2018, Camenisch et al. 2014). In the historical context, for example, water-driven mills played a key role in food security. Due to a lack of water, horse mills or hand mills were set into operation. In today's context, on the other hand, 460 the lower energy production of hydropower plants, or the shutdown of nuclear power plants due to low water or high water temperatures plays a major role (DWD 1947a,b, BNN 1947, DWD 1949a,b, BNN 1949, Erfurt et al. 2019). In addition, within the historical period, i.e. before the establishment of the official measuring network in 1881, the social and technical possibilities and structures changed. From 1800 onward, the dominant agricultural-feudal structures were gradually replaced by 465 the industrial revolution, also involving new possibilities for adaptation, e.g. improved infrastructure, better technical equipment such as pumps, more expertise and better hygiene and social welfare measures. The striking emigration waves of the 19th century from Germany to North America, which were also triggered by drought events, can be seen as another new adaptation option (Glaser et al. 2017). Additional innovations were the advent of the insurance industry (Kiermayr-Bühn 2009) or 470 institutional reactions in the modern age, understood as governance. Examples are the DWD's establishment of the heat warning system in 2005 as a result of the extremely hot summer of 2003 (Matzarakis 2016) with its unexpectedly large number of heat deaths or the implementation of a new 2019 drought index after the extreme year of 2018 (DWD 2019).

Nonetheless, there are many similarities between the recent and historical pathways. This enables 475 the direct comparison of historical derived indices and classified descriptions with current assessments of drought categories and classifications according to NDMC (2018) and the German Weather Service (DWD 2018, 2019).

The comprehensive data collections and derived time series also enable to identify outstanding and correspondingly well-documented extreme drought events, as given in the one-year filtered time 480 series (see Fig.8 and Tab. 4)).  Such compilations and rankings are quite common (f.e. Noone et al. 2017), but differ, depending on the selected parameters and chosen indicators.

One well known example is the extreme drought of 1540 (Wetter et al. 2014). The classification of 1540, its intensity and duration caused a longer and controversial discussion between the Dendroclimatological Community and the Historical Climatology Community.  With the presented 485 approach we can underline the outstanding drought situation of 1540. Other ones likewise the extremes of 1522, 1590, 1615, 1706 already had been described by Glaser (2013). The same with the

exceptional drought during the 19th century like 1834 by Erfurt et al. (2019) and 1842, which had been analysed by Brazdil et al. (2019) in the european context. The specific situation of 1921 also had been outlined by (Erfurt et al. 2019 and Brooks & Glasspoole 1922), as well as the more recent of
2003 described by Hémon et al. 2003 and Poumadère 2005 for France and Koppe & Jendritzky (2014), Glaser et al. 2018 and Erfurt et al. (2019) for Germany. Some identified droughts had their focus in other regions, like 1503 in Hungary (Kiss et al. 2017) or 1976 in Great Britain (Doornkamp et al. 1980). Overall, such rankings are widespread. Depending on the weighting and methodological approaches, the drought catalogs rankings vary somewhat for each year. In this sense a quite
reasonable number of identified single drought events can be cross validated by existing research results, while other ones likewise the identified drought of 1893 can be added. Only contemporary papers point to this extreme year for England and parts of the Continent (Lowe 1893, Brodie 1894).

The comparison of the identified extremes with the ones modelled by Cook et al. (2015) also shows the specific effects of trees. While f.e. 1615 is indicated by a longer drought from the written
documents, the dendro signal peaks first in the following year 1616. The same can be stated for 1834/1835 (see Tab. 4).

Increasingly, there is also the question of social impacts and long-term developments, especially regarding the effects of climate changes on human history. Such questions are usually answered using multiproxy approaches (Brazdil et al. 2019, Büntgen et al. 2010, 2011). In how far the
methodical and content-related insights presented in this study can be integrated into these questions will be the subject of future analyses.

Historical reconstructions can contribute to the question, if the recent extreme droughts such as in 2018, 2015 and 2003 must be seen as another indicator of climate change. The given results show that the recent development of precipitation alone is still within the historical variability.

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

## Acknowledgements

This study was carried out within the interdisciplinary research project DRIeR. The project is supported by the Wassernetzwerk Baden-Württemberg (Water Research Network), which is funded by the
Ministerium für Wissenschaft, Forschung und Kunst Baden-Württemberg (Ministry of Science, Research and the Arts of the Land of Baden-Württemberg). The authors like to thank Mathilde Erfurt for support in providing the SPI data series for Germany (1881-2018, data source: DWD).


# Figures

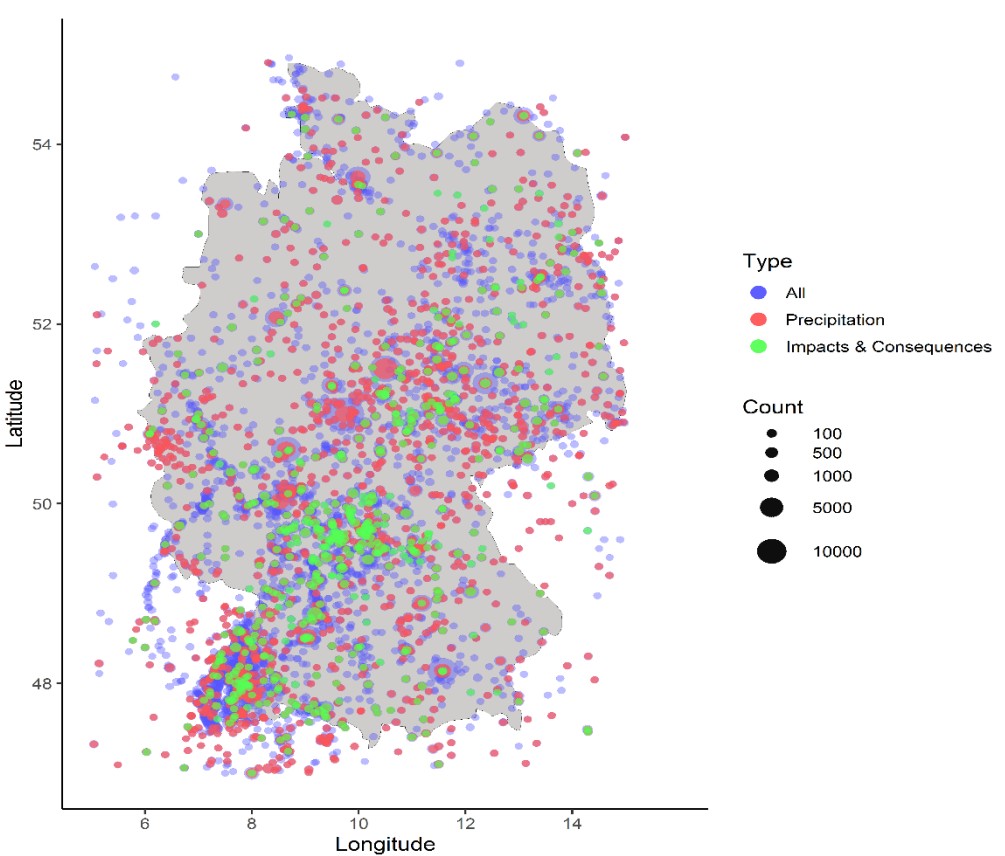

Fig. 1: Spatial distribution of all records referring to Central Europa in *tambora.org* since 1500 (blue), precipitation, dryness and drought records (red) and impacts and consequences (green).

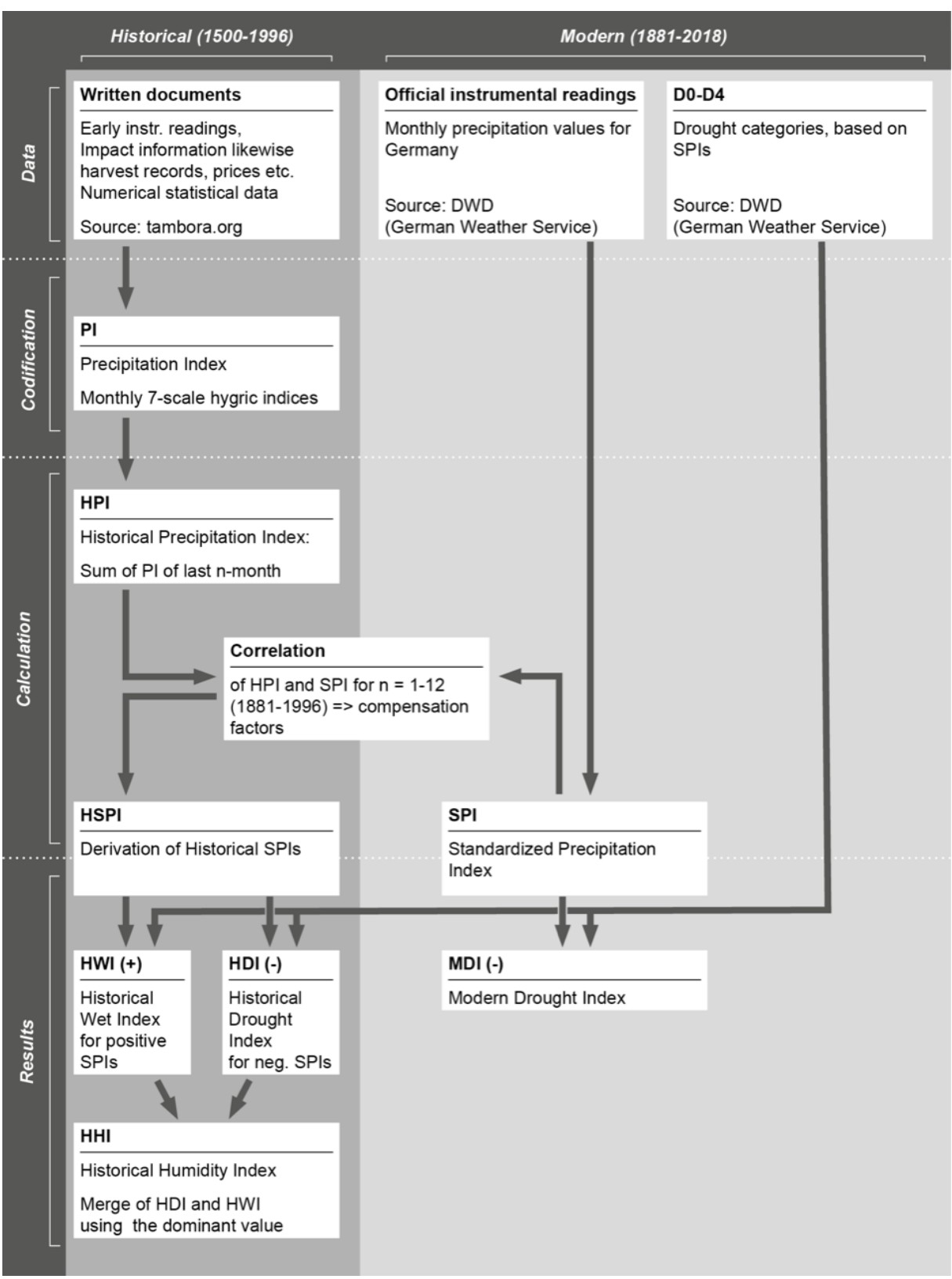

Fig. 2: Conceptual design of the analysis, illustrating the workflow and the single steps deriving the specific indices and their relations


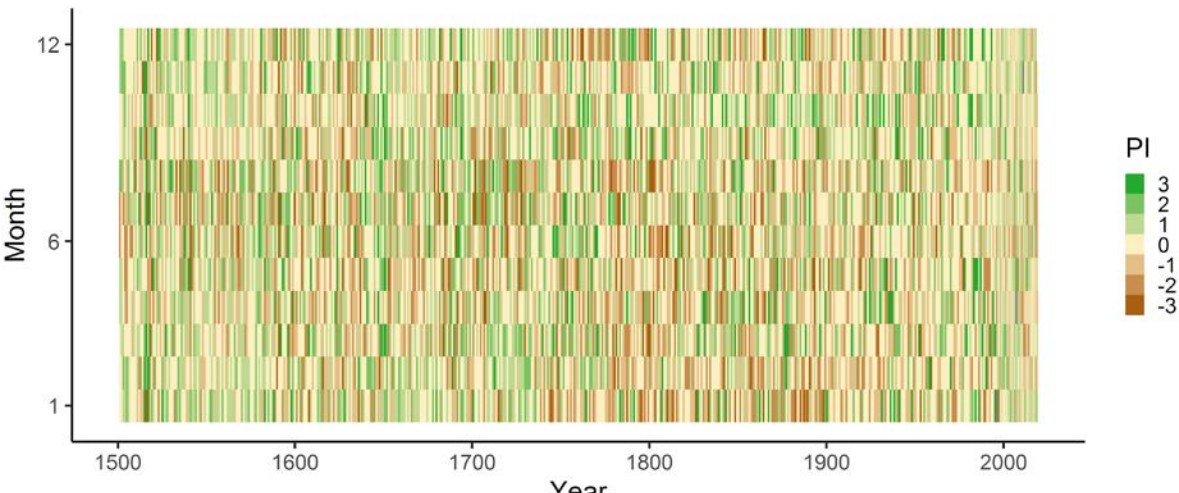

Fig. 3: Summary of the monthly precipitation index (PI) for Germany from AD 1500-2018

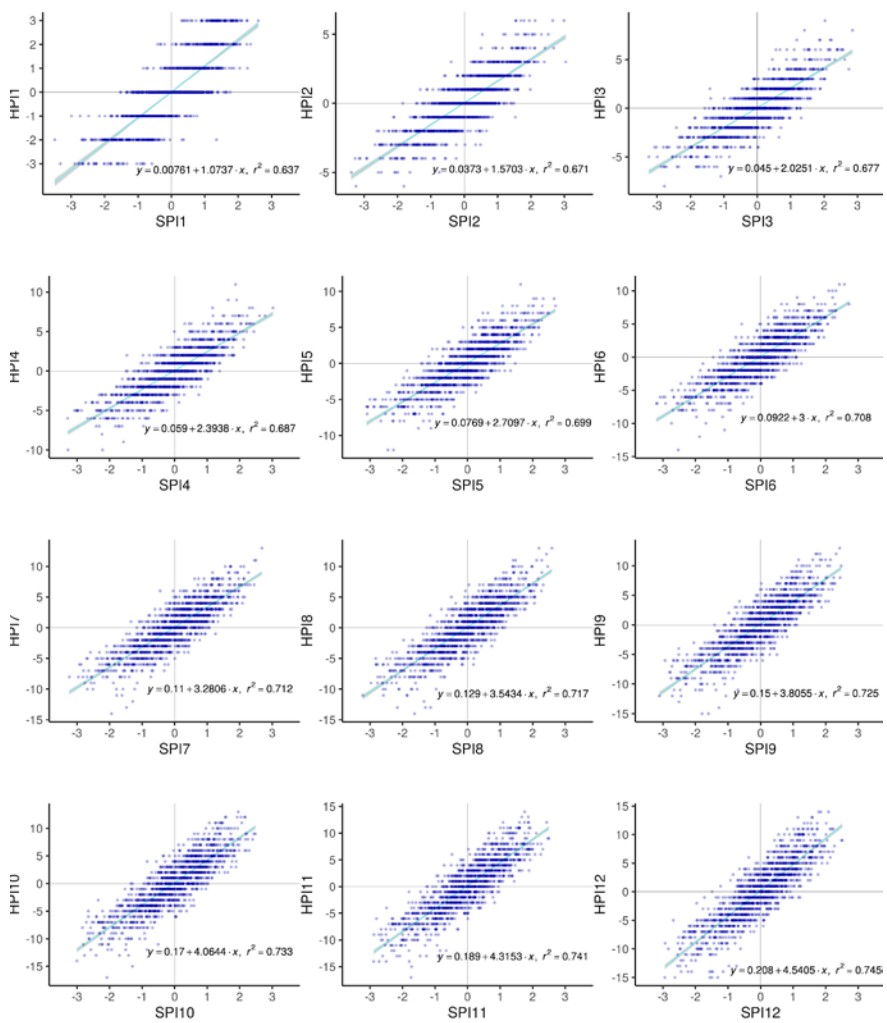

Fig. 4: Strength and shape of the relationship between SPI and HPI 1881-1996 for the duration of one to twelve months

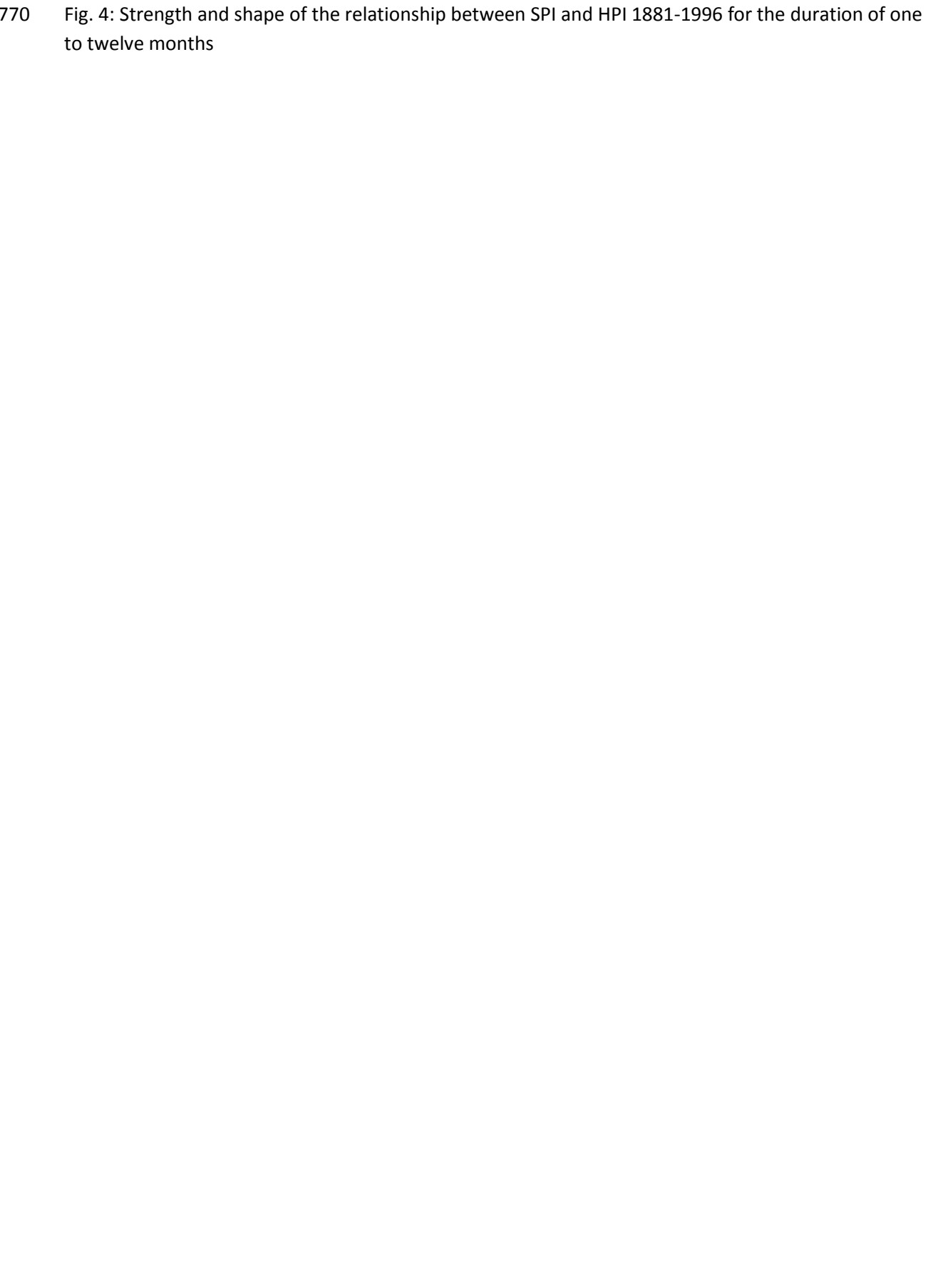

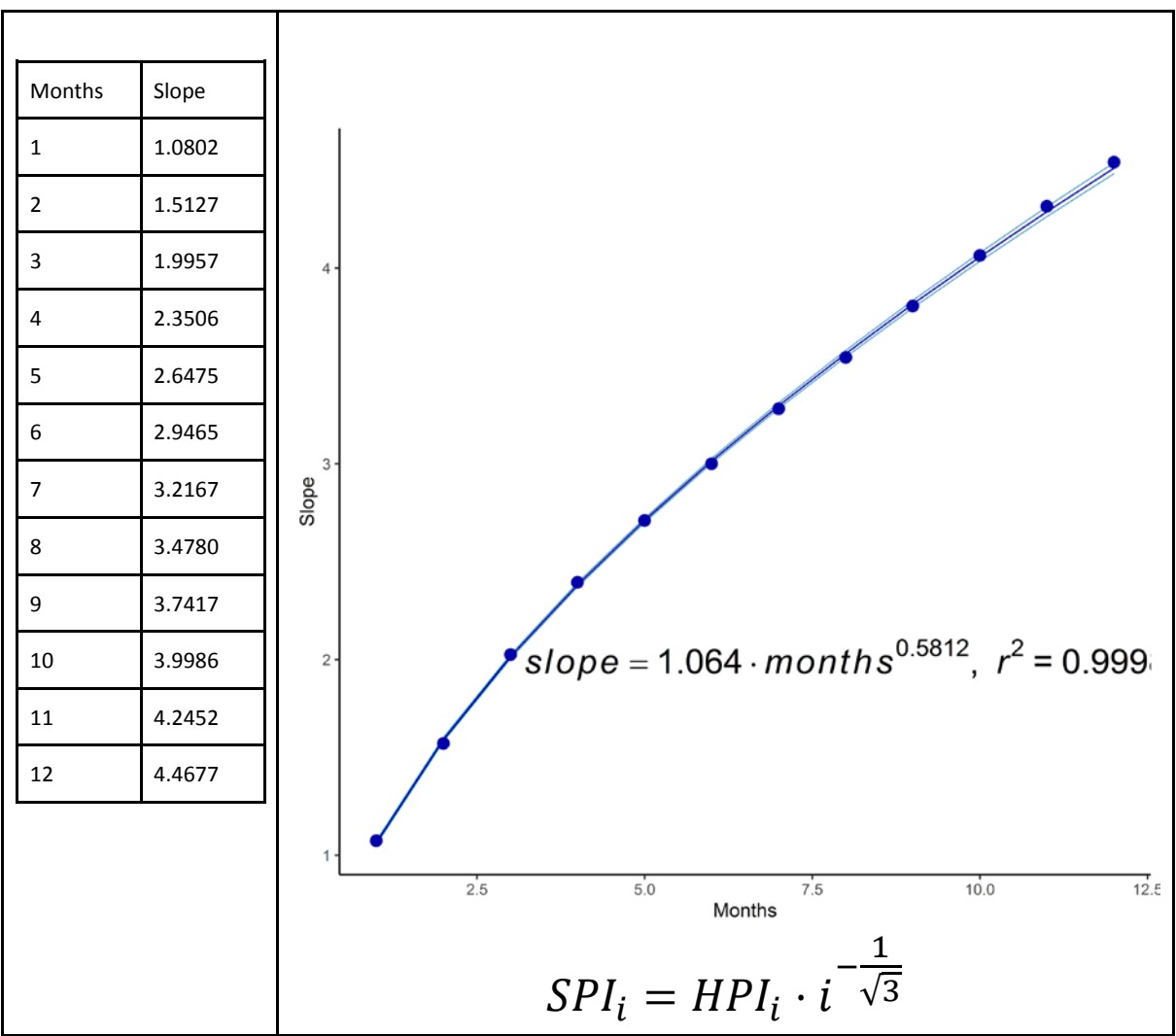

| Months | Slope |
|--------|--------|
| 1 | 1.0802 |
| 2 | 1.5127 |
| 3 | 1.9957 |
| 4 | 2.3506 |
| 5 | 2.6475 |
| 6 | 2.9465 |
| 7 | 3.2167 |
| 8 | 3.4780 |
| 9 | 3.7417 |
| 10 | 3.9986 |
| 11 | 4.2452 |
| 12 | 4.4677 |

$$slope = 1.064 \cdot months^{0.5812}, \; r^2 = 0.999$$

$$SPI_i = HPI_i \cdot i^{-\frac{1}{\sqrt{3}}}$$

Fig. 5: Duration and scale factors of the relationship between SPI and HPI for the duration of one to twelve months


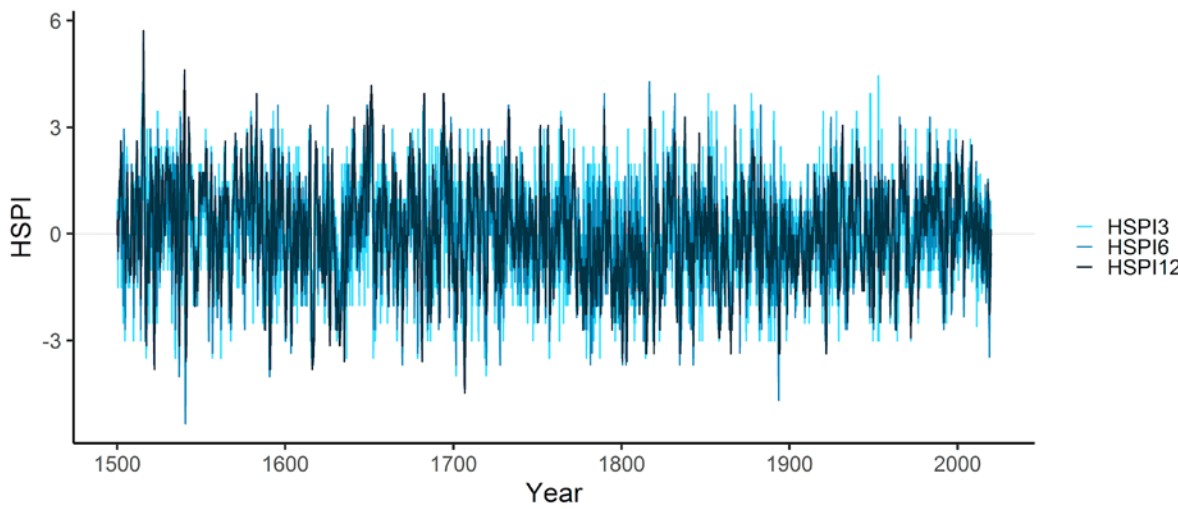

Fig. 6: HSPI3, HSPI6 and HSPI12 for Germany since 1500 for three, six and twelve months periods

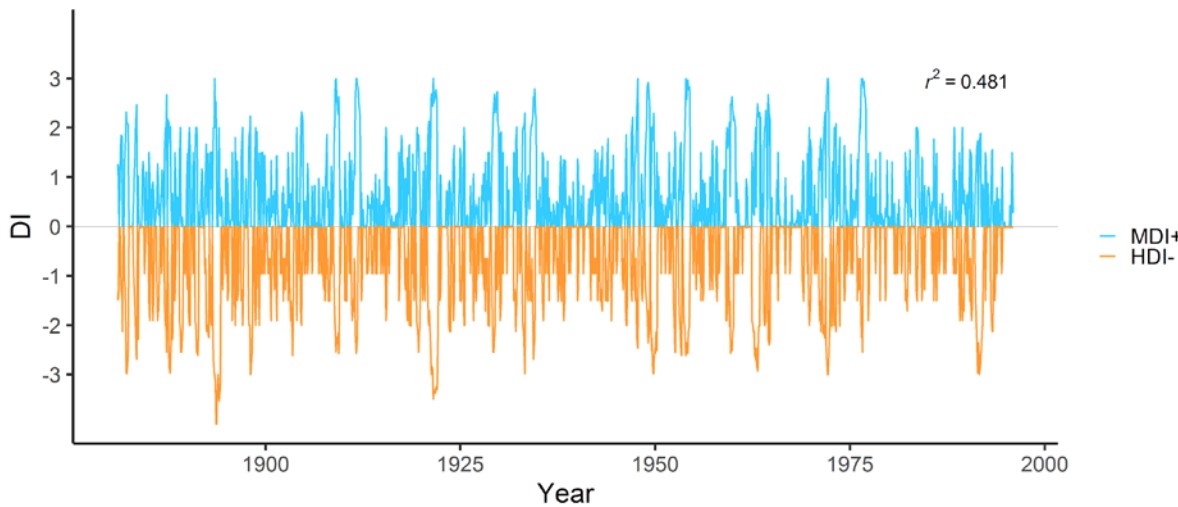

Fig. 7: Comparison of the Historical Drought Index (HDI) and the Modern Drought Index (MDI) for
Germany 1881-1991

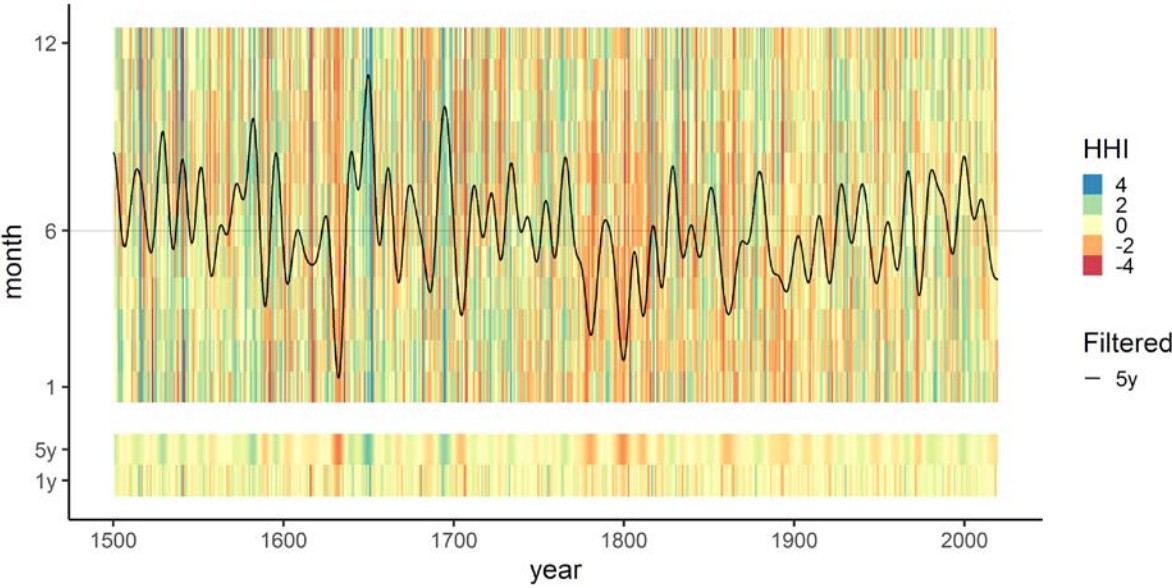

Fig. 8: Historical Humidity Index (HHI) for Germany since 1500. The upper part represents the monthly HHI from January to December for each year and the overlaid filtered five-year low pass (black line). The lower part indicates the low-pass values represented as colored scheme for five years and one year

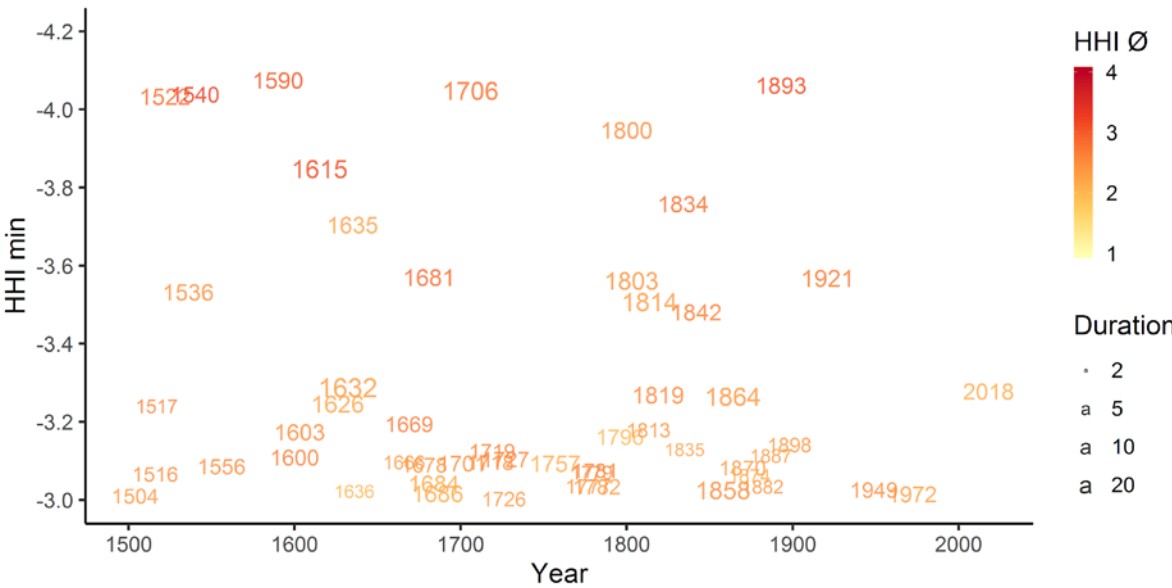

Fig. 9: "Yearcloud" of classified years since 1500. The average intensity is reflected by the color scheme and duration (month) by font size.
