# Peer review of "Reconstructions of Droughts in Germany since 1500 combining hermeneutic information and instrumental records in historical and modern perspectives"

_Climate of the Past, 2019_

## Referee Comment (RC1) · Anonymous Referee #1 · 12 Oct 2019

The manuscript "Reconstructions of droughts in Germany since 1500" focuses on the calculation of various indicators for droughts since 1500 based on the historical climate and environmental database tambors.org. Specifically a Historical Precipitation Index (HPI) is calculated and correlated with the SPI index. Additionally, a Historical Drought Index (HDI) and a Historical Wet index (HWI) are derived. Information on the long term development and dynamics of droughts is scarce and consistent long time-series are hardly available. However, the analyses of drought time series is highly relevant and important in the context of climate change and its impacts. For the development of sustainable risk management strategies for droughts it is important to know how droughts developed over time, and which drivers influenced their temporal dynamics to draw conclusions for the future. Thus, the research question dealt with and the objective of

the manuscript are innovative and highly relevant. However, I see the following major problems which require substantial re-writing of large parts of the manuscript as well as additional analyses during the revision. Thus, I suggest major revision (or even reject with an invitation to re-submit):

1) The development, i.e. calculation of all presented indicators is not provided. It therefore remains rather unclear what their specific meaning and their advantages and disadvantages are. It would be interesting to know, if all indicators are based on the seven-level monthly indicators for temperature and precipitation, which are included in the tambors.org database from 1500 onwards, or if additional data and information included in the database has been used. The development of the Historical Precipitation Index seems to be the key result of the manuscript, however, its development is described in one sentence only (lines 161-163). More detailed information is necessary here. The advantages and disadvantages of the developed indicators should be discussed.

2) I don't think that the droughts always affected whole of Germany in the same way. Thus, I doubt, that the indicators are continuously representative for whole of Germany. However, no information is given on spatial situation, how many stations are included in the calculation of the SPI, how are these distributed, how was the spatial aggregation undertaken? How representative for Germany is the HPI? How was spatial distributed information dealt with?

3) It is additionally unclear for what the presented drought indicators can be used and what analyses can be based on the developed time series. The calculation of these indicators should be complemented with analyses of the time series and their interpretation. It is quite strange, that the results and discussion section is rather a review and quite descriptive text only loosely connected with the indicators described in the methods section. I suggest to re-write the results section completely. It should present analyses of the developed indicators and time-series and their interpretation.

4) Also the outlook section is only very loosely connected to the rest of the manuscript. I suggest to completely re-write this chapter. It should rather contain ideas of how to further analyze the developed indicators and/or how these can be analyzed in combination with other drought information. Maybe, a conclusions section would be more relevant.

Further comments:

Introduction

Lines 20-23: These statements should be underpinned with references.

Lines 28-30: Statements could be more specific. Long-term reconstructions of what? Why are long term reconstructions necessary from comprehensive risk assessments? Century long time-series are not really necessary for comprehensive risk assessments, rather for temporally dynamic risk assessments.

Lines 41-46: the statements should be underpinned with references.

Lines 47-53: It would be interesting to know, which drought indices are characterising which drought type. If this is introduced in the introduction, this could be picked up later on for the new indices presented in the study, so that it becomes more clear for what the different indicators can be used.

Line 65: Please clarify to which phases you refer here. This is rather unclear.

Data

It is not fully clear to me, if all information/data described in this section is used for the analyses of the manuscript, maybe some more information which data/information has been used for what might be helpful.

Lines 98-101 This paragraph fits better into the introduction.

Lines 117-118. The equation for the calculation of the SPI should be provided or at

least described in more detail how it has been calculated and on which data specifically. "Official precipitation values for Germany" is not specific. How many stations? Daily values? Spatial aggregation etc. Or was the SPI calculated already and available in the database. This becomes not fully clear here.

Line 182-183: It remains unclear how the monthly PIs were summed up to the HPIs und how these were transformed into SPIs.

Is the MDI available in the database tambors.org?

Lines 188-190 it is not clear to me how the HDI was calculated on basis of the HPI. Please elaborate on this and explain better.

––––––––––––––––––––––––––––––––––––

---

## Referee Comment (RC2) · Anonymous Referee #2 · 3 Dec 2019

This manuscript is a general approach to study of drougths in historical dimension, using a large database. Different indices are implemented with overlapping to instrumental data period with more complete indices availability.

Historical dimension of drought is faced with a correct approach, considering it's a complex phenomena not easy to identify and evaluate in historical time, where not all information already is available for researchers. Justification of research is also well focused, with scientific and social preoccupation becase of increasing frequencies and severities of present drought events.

Definition of drougths. Authors describe from a general and integrated point of view. Avoiding conceptual problems. Correct references, and historical approach, where conceptual definitions are not so easy. A complete conceptual development could take

too pages. Context of manuscript, working on historical dimension, don't justify so detailed conceptual analysis.

Use of large database avoid the massive reference of sources and data previously available for this analysis. Bibliography is updated and complete. Absolutely adjusted to proposed research. Figures are well displayed and helps to understand results of manuscript. This manuscript, is a first analysis to show potential developments of historical droughts using quantitative and quantified information.

GENERAL ASPECTS + Title is too short. A subtitle could complete definition of proposed analysis.

+ Table 1. Very interesting proposal. Putting in relation drought duration with drought severity seems logic and useful to study drought in historical time, where information about definition and development of indices is not so complete and detailed as we would like. But, just a question about it. For a large natural region, as Germany or Central Europe, proposed table of criteria of classification is enough? Area under study is enoughly coherent or homogeneous to use only one system of criteria? Authors consider it would be possible application of similar method to be applied in different natural regions? Have they explored on this matter? Extension of this method to a larger spatial scale would be a good research path. Potential application of this method in other regions seems very useful. Authors could suggest any consideration about it?

+ Concerning method proposed for indexing drought phenomena, manuscript show a single construction of index. Related with previous questions. All information available for Germany is reduced to one index with proposed method. Authors consider this only index is representative of drought variability for all Germany? On the other hand, it exist any wheighting process or statistical method to generate this index considering different climatic contexts? All information is considered in a similar way or level? Any clarification about it would be useful.

+ Line 206. "The consequences and impacts (of drought) on the environment and

society can also be reconstructued very well". This matter has increasing interest. Integrated approaches for natural and social dimension of hydroclimatic extremes. But authors only mention this potential in one sentence. It could be possible additional description of these potentialities, under point of view of authors? Sources, density and diversity of available information... For example, what oppinion about complementary sources, as economic (taxation records, tithes, market oscillations of prices... or other related aspects, as records about water resources. Any consideration about this dimension of drought impacts, would be interesting to reinforce sentence of line 206.

+ Section "Outstanding single years". Lines 244-252. A more detailed description or analysis was expected. A short relation of years with drought, not chronologically ordered, with no clear explanation about severity or duration of respective drought characteristics. Please, could you explain into text what characteristics or reasons justify for every date singularity of drought recorded? Why these years are "outstanding"?. What they have in common? Any figure about characteristics of singularity: duration? Extension?, severity? any combination of magnitudes? Considering important dimension of database tambora.org, manuscript could include a more detailed analysis about extraordinary drought events? It would be an excellent opportunity to exchange knwoledge of these events to other colleagues, promoting comparative analysis in different spatiotemporal scales.

SPECIFIC ASPECTS

+ Lines 128-129, 152, 214, 239. Definition and use of concept of "cascade effects" (as impacts of droughts). Term is clear, but it could be improved with a more adjusted concept? Could be possible change "cascade" by "cumulative" effects? In fact, a cascade is water flowing downstream, meanwhile impacts of drought are increasing by addition in the same place. On the other hand, use of water-related phenomena, when drought is an important absence/shortage of water..... it seems even ironic!

Line 166. Exclamation sign. Better final point.

Lines 167-168. Unclear. Please, complete or clarify sentences.

Line 172. Formula doesn't appear clearly showed in text. May be by any editing problem. A black dot covers partially final part of formula.

Line 265. "Prominet" by "prominent"

————————————————————

---

## Author Comment (AC2) · 17 Dec 2019

We thank the rev.2 for the helpful comments and corrections very much!

This manuscript is a general approach to study of drougths in historical dimension, using a large database. Different indices are implemented with overlapping to instrumental data period with more complete indices availability. Historical dimension of drought is faced with a correct approach, considering it's a complex phenomena not easy to identify and evaluate in historical time, where not all information already is available for researchers. Justification of research is also well focused, with scientific and social preoccupation becase of increasing frequencies and severities of present drought events. Definition of drougths. Authors describe from a general and integrated point

of view. Avoiding conceptual problems. Correct references, and historical approach, where conceptual definitions are not so easy. A complete conceptual development could take too pages. Context of manuscript, working on historical dimension, don't justify so detailed conceptual analysis. Use of large database avoid the massive reference of sources and data previously available for this analysis. Bibliography is updated and complete. Absolutely adjusted to proposed research. Figures are well displayed and helps to understand results of manuscript. This manuscript, is a first analysis to show potential developments of historical droughts using quantitative and quantified information.

GENERAL ASPECTS + Title is too short. A subtitle could complete definition of proposed analysis.

Answer: Right. We have now a longer subtitle added

Table 1. Very interesting proposal. Putting in relation drought duration with drought severity seems logic and useful to study drought in historical time, where information about definition and development of indices is not so complete and detailed as we would like. But, just a question about it. For a large natural region, as Germany or Central Europe, proposed table of criteria of classification is enough? Area under study is enoughly coherent or homogeneous to use only one system of criteria? Authors consider it would be possible application of similar method to be applied in different natural regions? Have they explored on this matter? Extension of this method to a larger spatial scale would be a good research path. Potential application of this method in other regions seems very useful. Authors could suggest any consideration about it?

Answer: This is indeed a crucial question, we discussed within the author team several times – there are different natural landscape types- true – But we decided to adjust our approach to the existing modern DWD concept, in which the whole of Germany is represented by one indicator as pointed out in table 1. It is a question of spatial scaling.

Concerning method proposed for indexing drought phenomena, manuscript show a

single construction of index. Related with previous questions. All information available for Germany is reduced to one index with proposed method. Authors consider this only index is representative of drought variability for all Germany? On the other hand, it exist any wheighting process or statistical method to generate this index considering different climatic contexts? All information is considered in a similar way or level? Any clarification about it would be useful.

Answer: As pointed out in the previous paragraph, we adjusted our approach to the modern existing drought index – to find a comparable one dimensional representation. But analyzed and used a highly resoluted monthly time scale. The focus on the time means, that we could refer to the duration and the strength which is necessary to derive the chain of effects and the socio-economic dimension. The method itself is transferable, whenever SPI makes sense for the relevant regions - the results and conclusions in this article refer to modern Germany.

Line 206. "The consequences and impacts (of drought) on the environment and society can also be reconstructued very well". This matter has increasing interest. In-tegrated approaches for natural and social dimension of hydroclimatic extremes. But authors only mention this potential in one sentence. It could be possible additional description of these potentialities, under point of view of authors? Sources, density and diversity of available information... For example, what oppinion about complementary sources, as economic (taxation records, tithes, market oscillations of prices... or other related aspects, as records about water resources. Any consideration about this dimension of drought impacts, would be interesting to reinforce sentence of line 206.

Answer: We have now a differentiated explanation in paragraph 3.2. where we describe the historical pathways and drought categories in detail, especially the chain of effects.

Section "Outstanding single years". Lines 244-252. A more detailed description or analysis was expected. A short relation of years with drought, not chronologically or-dered, with no clear explanation about severity or duration of respective drought characteristics. Please, could you explain into text what characteristics or reasons justify for every date singularity of drought recorded? Why these years are "outstanding"?. What they have in common? Any figure about characteristics of singularity: duration? Extension?, severity? any combination of magnitudes? Considering important dimension of database tambora.org, manuscript could include a more detailed analysis about extraordinary drought events? It would be an excellent opportunity to exchange knowledge of these events to other colleagues, promoting comparative analysis in different spatiotemporal scales.

Answer: We reorganized this paragraph, explained the character of single years more detailed and even added a new derived view graph for better understanding and visualization. There is also now a table containing the list of extreme events as derived using the different indices.

SPECIFIC ASPECTS+ Lines 128-129, 152, 214, 239.

Definition and use of concept of "cascade effects"(as impacts of droughts). Term is clear, but it could be improved with a more adjusted concept? Could be possible change "cascade" by "cumulative" effects? In fact, acascade is water flowing downstream, meanwhile impacts of drought are increasing byaddition in the same place. On the other hand, use of water-related phenomena, whendrought is an important absence/shortage of water..... it seems even ironic!

Answer: This is indeed very funny, but an often cited concept, because the duration of dryness is leading to different drought types (agrarian, hydrological and socio-.economic). We also used the term "Chain of effects" as well as "pathway", which makes it somewhat clearer.

Line 166. Exclamation sign. Better final point. Answer: Absolutely right. Done.

Lines 167-168. Unclear. Please, complete or clarify sentences. Answer: We reworked these sentences – and made it clearer and more understandable.

Line 172. Formula doesn't appear clearly showed in text. May be by any editing prob-lem. A black dot covers partially final part of formula. Answer: This is indeed an editing problem, which does not appear in the original MS. Line 265. "Prominet" by "prominent" Answer: We corrected and rewrote the whole paragraph.

Please also note the supplement to this comment:
https://www.clim-past-discuss.net/cp-2019-104/cp-2019-104-AC2-supplement.pdf

―――――――――――――――――――

[Figure]

**Fig. 1.** conceptual design

**Fig. 2.** year cloud

**Supplement:**

Tab. 2: Table of outstanding droughts since 1500 for Germany

| | Longer context of dryness | | Selected Indices | | | Reference based on treerings (after Cook et al. 2015) |
|---|---|---|---|---|---|---|
| Year | Start | End | HHI min | HSPI.6 | HSPI.12 | Cook DE scPDSI |
| 1503 | Apr 1503 | Aug 1503 | -2.56 | -2.03 | -0.26 | -5.54 |
| 1522 | Apr 1521 | May 1522 | **-4.00** | -3.69 | -3.81 | -0.17 (SE: -1.4) |
| 1540 | May 1540 | Mar 1541 | **-4.00** | -5.35 | -3.59 | -2.94 |
| 1567 | Apr 1566 | Jan 1568 | -2.99 | -2.36 | -2.48 | -2.13 |
| 1590 | Mar 1590 | Mar 1591 | **-4.00** | -4.02 | -3.81 | -3.03 |
| 1615 | Jan 1615 | Mar 1617 | -3.81 | -3.69 | -3.81 | -1.91 (1616: -3.79) |
| 1632 | Mar 1630 | May 1633 | -3.31 | -2.69 | -3.15 | -1.71 |
| 1635 | Mar 1634 | Apr 1635 | -3.78 | -3.35 | -3.59 | -3.36 |
| 1669 | May 1669 | Feb 1670 | -3.15 | -3.69 | -3.15 | -3.30 |
| 1681 | Jul 1680 | Sep 1681 | -3.59 | -3.03 | -3.59 | -3.13 |
| 1706 | Jun 1705 | Jul 1707 | **-4.00** | -4.35 | -4.47 | -2.07 |
| 1719 | May 1719 | Dec 1719 | -3.00 | -3.69 | -2.26 | -3.81 |
| 1800 | Nov 1799 | Jan 1801 | **-3.98** | -3.69 | -3.58 | -3.06 |
| 1803 | Mar 1802 | Oct 1803 | -3.59 | -3.69 | -3.58 | -3.73 |
| 1814 | Dec 1813 | Jan 1816 | -3.54 | -3.36 | -3.37 | -0.51 (N: -2.4) |
| 1834 | Feb 1834 | Feb 1835 | -3.73 | -3.69 | -3.37 | -2.76 (1835: -4.44) |

| 1842 | Jan 1842 | Jan 1843 | -3.49 | -3.69 | -3.15 | -3.06 |
|------|----------|----------|-------|-------|-------|-------|
| 1858 | Feb 1857 | Oct 1858 | -3.00 | -2.69 | -2.92 | -4.64 |
| 1864 | Dec 1863 | Jan 1866 | -3.37 | -2.36 | -3.37 | -2.53 |
| 1893 | Mar 1893 | Mar 1894 | **-4.00** | -4.69 | -3.37 | -4.17 |
| 1921 | Oct 1920 | Feb 1922 | -3.49 | -3.36 | -3.37 | -5.57 |
| 1947 | May 1947 | Oct 1947 | -2.46 | -2.36 | -2.04 | -3.96 |
| 1949 | Jun 1949 | Mar 1950 | -2.99 | -2.69 | -2.48 | -1.6 |
| 1963 | Jun 1962 | Jun 1963 | -2.92 | -2.69 | -2.70 | +0.37 |
| 1976 | Mar 1976 | Aug 1976 | -2.56 | -2.36 | -1.82 | -4.06 |
| 2003 | Mar 2003 | Dec 2003 | -2.47 | -2.33 | -1.76 | -1.36 |
| 2018 | Feb 2018 | Feb 2019 | -3.35 | -3.46 | -2.26 | - |

---

## Author Response (AR1)

[revised manuscript text omitted]

The information covers the German area well (Fig.1). All written sources were processed according to the methods used in historical climatology (Pfister 1999, Brazdil et al., 2005, Glaser 2013).

[Fig. 1: Spatial distribution of information in tambora.org]

All records in tambora.org are numerically coded, comprising spatial, temporal and content aspects. In addition to the coded events, the original text quotes are also included in the database so that the overall context and the coding can be retraced for each record. Revised and supplemented precipitation indices are available from 1500 to 1995 (Glaser 2013, Glaser & Kahle 2019).

Additionally, there are early precipitation measurements and derived drought indices from 1800 onward (Erfurt et al. 2019).

The information in tambora.org is coded, especially the spatial, temporal and content aspects are classified into numerical codes. The information relevant for the analysis of dryness and drought events, in particular the temperature and precipitation information are completely available from 1500 onwards as seven-level monthly indices (Fig. 2). In addition, hydrological extremes, in particular high and low water as well as phenological hints are available. In addition to the drought events there are many descriptions of the impacts on the environment and society. In addition to the coded events, the original text quotes are also included in the database so that the overall context and the coding can be traced at any time. The sources of drought are extremely varied and differentiated. For example, in the 123 sources for the Drought Year 1540, 41% of the data refer to agriculture, 17% to water, 11% to health, 10% to forest fires, 8% to soil and 8% to the environment and ecosystem. Basically, outstanding drought events are documented by particularly many and differentiated sources. Particularly well-documented events of the century include 1540, 1503 and 1534, 1615 and 1616, 1669 and 1684, and 1718 and 1719, 1834, 1842, 1865 and 1893, 1921, 1949, 1959, 1976, 2003 and 2018.

[Fig. 1: Spatial distribution of all records referring to Central Europa in *tambora.org* since 1500 (blue), precipitation, dryness and drought records (red), and impacts and consequences (green).]

The data relevant for the analysis are consistently available from 1500 onwards as classified monthly hygric indices using a seven-level scale (PI) (Fig. 3).

The monthly PI reveals a differentiated picture of drier and wetter periods since 1500.

In general, a particularly large number and higher differentiation of sources document outstanding drought events. One example is the drought year 1540, where 41% of the 123 sources refer to agriculture, 17% to water, 11% to health, 10% to forest fires, 8% to soil and 8% to environmental and ecosystem issues.  Other ouitstanding drought events are in the 16th century the droughts of 1534, 1536, 1540 and 1590.  In the 17th century 1615, 1616, 1632, 1635, 1669, 1684 and 1685 had been described as exeptional dry. The same for 1718, 1719, 1742 and 1749 during the 18th century. While in the 19th century, the years 1834, 1842, 1858, 1865 and 1893 are reported as drought years. For the 20th century this was the case in 1921, 1949, 1959 and 1976 and finally in the 21st century in 2003 and 2018.

The second, modern data set used for the analysis consists of the official precipitation data for Germany from 1881 onward. These values, recorded, averaged and provided by the DWD (Deutscher Wetter Dienst, the official German Weather Service) from the national official network stations, respresent the area of modern-day Germany. This study also draws upon the official drought categories D0-D4 and their definitions by the DWD (2019).

In addition, from 1881 onward the official precipitation data for Germany can be used. From 1800 onward early precipitation measurements and derived drought indices are available (Erfurt et al. 2019) as well as the revised and supplemented precipitation indices from 1500 to 1995 (Glaser 2013, Glaser & Kahle 2019).

[Fig. 2: Summary of the monthly precipitation index (PI) for Germany from AD 1500-2018]

**3 Methods**

The conceptual design of the analysis is given in Fig.2. It illustrates the single steps and the workflow as a whole. Each individual step is described in the following subchapters in detail.

[Fig. 2: Conceptual design of the analysis, illustrating the workflow and the single steps deriving the specific indices and their relations]

**3.1 Derivation of the monthly Precipitation Index (PI) since 1500**

The hygric indices (PI) were derived from the written evidence of the *tambora* sources via semantic profiles, a method well established in historical climatology (Glaser 1991, 1996, 2013, Glaser & Riemann 2009, Pfister 1999, Brazdil et al., 2005). Therefore, direct hygric indications as well as the descriptions of impacts and consequences are hierarchically ordered according to their intensity and assigned to the appropriate index value. A seven-scale index scheme, ranging from -3 to +3 with index 0 representing the average situation, has proven to be appropriate for the classification of historical records (Glaser 1991, 1996, Glaser & Riemann 2009, Glaser 2013).

Direct descriptions mostly refer to the absence of rain or the corresponding lack of clouds. In many cases, the duration is also indicated. The documents related to specific consequences describe effects on harvest results, the phaenological and ecological situation, but also hydrological consequences and impacts on economy, society, and their reactions. This correlates very well with

the definitions of meteorological, agrarian, hydrological, groundwater and socio-economic drought in modern classifications (NDMC 2018).

Generally, extreme events are represented by a larger number of sources in historical documents. Such information also spans wider areas, especially if the records refer to droughts. Additionally, the information is more detailed and quite often severe events are compared to previous ones. Such long-term memories persist across generations.

The hierarchical class assignment and its typical indicators for the precipitation indices (PI) -1 to -3 are presented as follows:

Index -1 is indicated by descriptions of a beginning rainfall deficit. There are often indications of higher damages relating to the harvest of rain-sensitive products such as hay, vegetables and other garden products.

Index -2 relates to a longer duration of lack of precipitation, prolonged heat and dryness. Average crop losses for main crops are reported as well as low water levels in smaller bodies of water and reduced spring fills. Heat stress on plants, premature leaf discoloration and the death of plant parts are observed, also dry cracks in soils, occasional forest fires and the impairment of infrastructure, for example related to shipping and water mills.

Index -3 represents extreme dryness revealing a chain of effects: After a prolonged period of dryness and heat, the agrarian consequences include severe crop losses and even harvest failures as well as emergency slaughteries due to fodder shortages. If the dryness lasts for weeks, several months or even seasons, there are integrating effects that correpond to reports like low water levels in greater lakes, ponds and larger river systems as well as the drying up of springs and wells. In addition, reports of excessive water shortage and the appearance of "hunger stones" are common. Ecological impacts include a generally visible heat stress of the vegetation, premature leaf discoloration and the withering of plants. In addition, dry cracks in soils, dust veils, dust storms and effects of wind erosion are indicated. There are diverse descriptions of a shift of the phaenological phases, e.g. early flowering, ripening and harvest, but also expression like "wine of the century" indicate dry years. The same is true for reports of forest fires and fish kills. The impairment of infrastructure, especially the termination of shipping and the failure of mills are frequently mentioned socio-economic impacts. The direct consequences for human health and well-being are also documented, e. g. through indications of heat stress and death, increased death rates, the outbreak of epidemics, diseases and hunger crisis due to a lack of food. In addition, the reports include price increases and speculations.

Documented authorities´ reactions range from restrictions and regulations on water access or rationing to the declaration of a state of emergency. Societal reactions like supplications, processions, pilgrimages, increasing irrational explanations and interpretations are quite common. The sources also report begging, moving around in order to seek for food, riots and protests, theft, looting, robbery and social excesses. These integrating effects allow conclusions to the preceding months, and in many cases, the exact dates of meteorological droughts are indicated by the name day of saints.

The indexing process is similar to modern classifications and definitions of drought categories. Such modern drought catgories also take into account the descriptions of impacts and societal consequences and reactions (McKee 1993, NDMC 2018).

Weather diaries with daily records exist for 60 years from the period 1500 to 1800 also on precipitation days (Glaser 1996, Glaser 2013). These records are compared with modern precipitation data on a monthly scale, enabling a comparison of numerical rainfall data with the classified written evidences, which serves as an additional verification of the index levels.

For the numerical records since 1800, we used a classification scheme based on normal distribution, index „0" reflecting the monthly average with a plus/ minus 0.75 –fold standard deviation. Index „-1" ranges betwen -0.75 and -1.5 standard deviation, Index „-2" between -1.5 and -2.25 fold standard deviation and „-3" below -2.25 standard deviation. The positive indices refer to the appropriate positive ranges. The period 1951-1980 was chosen as reference period. These numerically derived indices were combined with the hermeneutically derived ones.
The positive hygric index correponds to the humid and wet situations and is derived in the same manner. The summary of the monthly PI for Germany from 1500 onward is given in Fig. 3. The data is available via Glaser & Kahle (2019).

[Fig. 3: Summary of the monthly precipitation index (PI) for Germany from AD 1500-2018]

**3.2 Derivation of historical pathways and drought categories**

The consequences and effects of drought on environment and society recorded in historical sources resemble the structure also observed by recent approaches (Glaser et al. 2016, Brazdil et al. 2016): A precipitation deficit is followed by 
[revised manuscript text omitted]
 1500. Many of them are well known and already described in the literature for Germany and neighboring regions of Central Europe like the droughts of 1540 (Wetter et al. 2014), 1590, 1622, 1631/32, 1706, 1719 (Glaser 2013), or the ones in 1834 and 1921 (Erfurt et al. 2019) or 1842 (Brazdil et al. 2019).

**4 Results, discussion and conclusion**

The article presents different index-based reconstructions of monthly and yearly drought time series for Germany since 1500. The reconstructions are based on historical records from the virtual research environment tambora.org as well as modern instrumental records. Written documents are also taken into account to analyse the climatic effects and consequences on environment and society. These chain of effects reflect very strongly the common accumulating effects of modern definitions of drought types. Therefor they are also comparable to modern drought categories (McKee 1993, NDMC 2018).

At first, a seven-scale index scheme is used to deduce a monthly Precipitation Index (PI) since 1500. This method is well established in historical climatology (Glaser 1991, 1996, 2013, Glaser & Riemann 2009, Pfister 1999, Brazdil et al., 2005). Hygric indications as well as the descriptions of impacts and consequences, here referred to as chain of effects are hierarchically ordered according to their intensity and assigned to the appropriate index value. The process is similar to modern classifications and definitions of droughts. In difference to modern drought-indices like SPI, SPEI or PDSI derived from modern instrumental records, these classifications directly take into account the descriptions of impacts and societal consequences and reactions (see also Erfurt et al 2019).

The derivation of the historical precipitation index (HPI) from the monthly precipitation indices (PI) - including the positive deviations - is the sum of the corresponding number of months of the period 1881-1996, analogous to the SPI. We calculated the HPI for time windows from one to twelve months in order to map the accumulative effects of dryness and lack of rainfall as well as for humid and wet conditions. The derived Historical Precipitation Index (HPI) is correlated with the Standardized Precipitation Index (SPI). A calibration factor had been calculated and applied to derive the Historical Standardized Precipitation Index (HSPI). In this sense, the HSPI reflects the longer rainfall deficits and dryness in a more comparable way to modern statistical approaches. Finally, a Historical Drought Index

(HDI) and a Historical Wet Index (HWI) are derived from the hygric indices seperately. Both are combined for the Historical Humidity Index (HHI).

[revised manuscript text omitted]

Nonetheless, there are many similarities between the recent and historical chain of effects. This justifies the parallelizat of past and historical drought categories,  combin hermeneutic criteria with empirically derived indices.

In Tab.1 we synthesized an evaluation scheme to relate historical descriptions with the corresponding modern descriptive ratings and selected numerically derived indices. This enables the direct comparison of historical derived indices and classified descriptions with current assessments of drought events according to NDMC (2018) and the German Weather Service (DWD 2018, 2019).

The comprehensive data collections and derived time series also enable to identify outstanding and correspondingly well-documented events of the centuries.

, as given in the one-year filtered time series. The written evidence often allows differentiated statements about the underlying climatic causes and in many cases information is available about the temporal structure, particularly the onset, the end and the course of droughts. They are thus similar in content to other Central European historical sources (Pfister 1999, Brazdil et al. 2019). The written evidence covers not only the climatic development but also the impacts and consequences as well as

the responses and adaptation strategies of societies. This allowed the application of the concept of pathway analysis, especially for extreme drought events, which in turn served as baseline for the derivation of the evaluation scheme (Tab.1).

The most outstanding drought events occurred in the 16th century in 1590 and 1540, but also in 1536. In the 17th century, this is the case in 1615 and 1616 and during the striking sequence 1630-1635, 1669 and 1685. Outstanding droughts occurred during the 18th century in 1706, 1718 and 1719, 1742, and 1749; in the 19th century in 1834, 1842, 1865 and 1893; in the 20th century in 1921, 1949 and 1959; and finally in the 21st century in 2003 and 2018. Many of these extremes are not only confirmed in other papers (Koppe & Jendritzky 2014, Cook et al. 2015, Glaser et al. 2018, Erfurt et al. 2019), but have also been identified in neighbouring 
[revised manuscript text omitted]

**Editor Decision: Reconsider after major revisions** (01 Feb 2020) by Günter Blöschl
Comments to the Author:
The two reviews provide useful suggestions for strengthening the paper. I consider the paper interesting but more methodological detail is needed in order for the reader to understand what has been done. There are also a number of inconsistencies. I recommend that the authors address all comments of the two reviewers. The responses to the reviews seem to be an appropriate path to follow.

Additionally I have the following observations:

1. More detail on the data is needed. What is 'information' exactly? Are these 280000 pieces of information all relating to droughts – how many places, years?

Actually there are 330.000 coded records in tambora.org for Central Europe as represented as blue dots in Fig. 2.. The 54.000 records, indicating precipitation and specific information regarding dryness and droughts are given in red. Additional 12.600 records, which describe the impacts and consequences of dryness, drought and lack of precipitation, likewise water shortages, low water levels of larger rivers, fish kills, forest fires, emergency slaughteries, crop failures, or prayer for rain etc. are given in green. All in all the information covers large parts of Central Europe. The southwest and the center as well as the eastern parts of Germany are particularly well depicted, but also the larger river systems such as the Main, Rhine and Elbe. The spatial distribution also refers to the cultural, political, economic and religious centers such as Nuremberg, Cologne, Leipzig, Erfurt, Hamburg and Mainz as well as other larger cities and monasteries. The coastline respectively the harbour locations likewise Hamburg, Lübeck and Rostock are also well represented.

The temporal coverage is very good, i.e. there has been information for every month since 1500. Basically, the average and inconspicuous months are less documented than more extreme ones. Periods for which daily, systematic information from weather diaries are available have correspondingly denser assignments (see also Glaser & Riemann 2009).

How did you get the monthly indices of Fig. 2?

The hygric indices had been derived from the written evidence of the sources via semantic profiles, a method well established in historical climatology (Glaser 1991, 1996, 2013, Glaser & Riemann 2009, Pfister 1999, Brazdil et al., 2005). In such semantic profiles, the direct hygric hints as well as the descriptions of the impacts and consequences are hierarchically ordered due to their intensity and assigned to the appropriate index value. The process is similar to modern classification and definitions of droughts, which beside different drought-indices likewise SPI, SPEI, PDSI oder scPDSI derived from modern instrumental records also take into account descriptions of impacts and societal consequences and reactions (McKee 1993, NDMC 2018).
A seven-scale index system, ranging from -3 to +3 has proven appropriate for the classification of historical records (Glaser 1991, 1996, Glaser & Riemann 2009, Glaser 2013).

Generally, extreme events are represented in historical documents by a larger number of sources. Such information is also spanning wider areas, especially if the records refer to droughts. Additionally the information is more detailed. Often severe events are compared to previous ones. Such long-term memory persisted across generations.

Direct descriptions mostly refer to the absence of rain or the corresponding sky coverage and the lack of clouds. In many cases, the duration is indicated, too. The documents related to the specific pathways and consequences describe the effects onto harvest results, the phaenological and ecological situation, but also the hydrological consequences and the impacts onto economy and society and their reactions. This correlates very well with the definitions of meteorological, agraian, hydrological, groundwater and socio-economic drought in modern classificatioons (NDMC 2018).

The evaluation scheme for the hygric-index is ranging from -3 to +3.

Index 0 represents the average situation.
Index -1 is indicated by description of deficits of rainfall. There are often indications of less impairment of the harvest of rain-sensitive products such as hay, vegetables and other garden products.
Index -2 increased number of indications of lack of precipitation, prolonged heat and dryness. Average crop losses for main crops are reported as well as low water levels in smaller bodies of water, reduced spring fills. Heat stress on plants, premature leaf discoloration, death of parts of plants is quite obvious, also dry cracks in soils, occasional forest fires. Impairment of infrastructure, for example shipping and water mills.
Index -3, representing extreme dryness revealing a chain of effects as follows:  Extreme dryness is indicated by large numbers of records of prolonged dryness and haet, caused by lack of rainfall. The agrarian consequences are severe crop losses and even harvest failures, emergency slaughteries due to lack of feed.
If the dryness lasts for weeks, several months and seasons, there are integrating effects that correponds to reports likewise low water levels in greater lakes, ponds and larger river systems as well as drying up of springs and wells. Also reports of excessive water shortage and the appearance of "hunger stones" are common. The ecological impacts refer to generally visible heat stress of the vegetation, premature leaf discoloration and the withering of plants. In addition, dry cracks in soils, dust veils and dust storms and effects of wind erosion are indicated. There are plentiful descriptions of clear premature of the phaenological phases likewise flowering, ripening and harvest, but also "wine of the century". Reports of forest fires and fish kills are also common. The impairment of infrastructure, especially the termination of shipping and the failure of mills are reported as socio-economic impacts. The direct consequences for human health and well-being are also documented: Indications of heat stress, heat death, and increased death rate, outbreak of epidemics, diseases and due to lack of food hunger crisis. The scenarios also include price increases and speculations.
Authorities reacted with restrictions and regulations on water access or rationing. Finally there are the adoption of special regulations and the declaration of a state of emergency.
Societal reactions as supplications, processions, pilgrimages, increasing irrational explanations and interpretations are quite common. Also begging, moving around for food seekers, riots and protests, theft, looting, robbery and social excesses.
These integrating effects allow conclusions to the preceding months. In many cases the exact date of such a meteorological drought is indicated by the name day of saints.
The positive hygric index correponds to the humid and wet situations and are derived in the same manner.

In addition, for 60 years oft he period 1500-1800, precipitation days are given from daily weather diary entries (Glaser 1996, Glaser 2013). Such high resoluted daily entries are compared with modern precipitation data on a monthly scale. This enables a comparison of numerical rainfall data with the classified written evidence on a monthly scale and can be regarded as an additional verification of the index levels.

For the numerical records since 1800, we used a classification scheme based on normal distribution. Index „0" reflects the monthly average with a plus/ minus 0.75 –fold standard deviation. Index „-1"

ranges betwen -0.75 and -1.5 standard deviation, Index „-2" between -1.5 and -2.25 fold standard deviation and „-3" below -2.25 standard deviation. The positive Indices refer to the appropriate positive ranges.  The period 1951-1980 was chosen as the reference period. These numerically derived indices were combined with the hermeneutically
derived ones.

 I would not expect there is information in every singly month of the past 500 years.

There are records for all months since 1500 due to the Gutenberg revolution (printing machineries, distribution of paper production as well as the increase in writing and reading abilities…), before 1500 this is not the case.

The derivation of PI is completely missing, yet essential for understanding the paper.
The derivation is now described in detail above. Also the data information.

2. CP has an open data policy. Please deposit data in a public repository
https://www.climate-of-the-past.net/about/data_policy.html

The data are already available via

https://www.re3data.org/search?query=tambora

and assets for code and data  are added to the article:

**Temperature and Hygric Indices for Central Europe since AD 1500** G. Rüdiger and M. Kahle
https://doi.org/10.6094/tambora.org/2019/c493/csv.zip

**climdata/drought2019: Reconstructions of Droughts in Germany since 1500** M. Kahle and R.Glaser
https://doi.org/10.5281/zenodo.3405167

3. Fig. 3 is difficult to understand (in fact the figure captions are generally too short). Is each point a year, averaged over Germany? And if so, how?

We enlarged the captions.

Each point represents a month and the average over Germany.

We added a new conceptual figure which clarifies the derivation and relations of the introduced indices very well.

4. Fig. 4 and the associated methods are unclear to me. They need a full description. Even though I do not know the details of the method, the correlation of 0.99 does not look right.

We reworked and reorganized the chapters and subchapters for each single step, so that – together with the new conceptual design - the calculations and their relations to each other should be clear.

0.99 is not the correlation of single months, but the correlation oft he relation bewteen SPI and HPI for durations of 1 to 12 months. It proofs, that summerizing PIs are equivalent to the calculation of SPIs.

5. Correlations HDI-MDI is r=0.478 in line 191 but r=0.75 in line 273.

The relevance of these correlations are difficult to assess unless the basis of the data becomes clearer. Have any systematic data been used in the historic data set? It would be good to demonstrate by a recent drought example how the documentation based data set compares with precipitation measurements.

You are absolutely right …the correlation coefficient is r=0.48

Please address the comments of the reviewers and my observations and submit a revised paper.

The manuscript "Reconstructions of droughts in Germany since 1500" focuses on the calculation of various indicators for droughts since 1500 based on the historical climate and environmental database tambora.org. Specifically a Historical Precipitation Index(HPI) is calculated and correlated with the SPI index. Additionally, a Historical Drought Index (HDI) and a Historical Wet index (HWI) are derived. Information on the long term development and dynamics of droughts is scarce and consistent long time-series are hardly available. However, the analyses of drought time series is highly relevant and important in the context of climate change and its impacts. For the development of sustainable risk management strategies for droughts it is important to know how droughts developed over time, and which drivers influenced their temporal dynamics to draw conclusions for the future. Thus, the research question dealt with and the objective of manuscript are innovative and highly relevant. However, I see the following major problems which require substantial re-writing of large parts of the manuscript as well as additional analyses during the revision. Thus, I suggest major revision (or even reject with an invitation to re-submit):

Answer: The authors thank rev1 for the general statements and comments! In the meantime, we revised the whole article along the suggestions; we also added a new conceptual viewgraph to highlight the analytical work-flow and reorganized the whole article along this concept.

1) The development, i.e. calculation of all presented indicators is not provided. It therefore remains rather unclear what their specific meaning and their advantages and disadvantages are.

Answer: We added a new view graph to highlight the whole workflow and especially to present the calculation of all presented indicators precisely (see Figure 2 in the revised MS). We reorganized the relevant paragraph and added the meaning and advantages of the used indicators. We referred to the well introduced modern indices (SPI, Drought Classes of the DWD), which are widely known and used. The historical derived Indices are related to these to get the opportunity to calibrate, connect and compare these.

It would be interesting to know, if all indicators are based on the seven-level monthly indicators for temperature and precipitation, which are included in the tambors.org database from 1500 onwards, or if additional data and information included in the database has been used.

Answer: We used the seven class hygric index from tambora.org, but also the given information (written evidence) of the impacts on agriculture, forestry, water balance, ecology and socio-economic effects.

The development of the Historical Precipitation Index seems to be the key result of the manuscript, however, its development is described in one sentence only (lines 161-163). More detailed information is necessary here. The advantages and disadvantages of the developed indicators should be discussed.

Answer: We see the HPI, which is equivalent to the modern SPI, as a key issue, but also as a stepping stone for the derivation of the HSPI (Historical SPIs). This is now much clearer in the new introduced concept figure. The whole structure of the workflow, which is described in the sub-paragraphs 3.1 to 3.9 is much clearer now and more informative.

2) I don't think that the droughts always affected whole of Germany in the same way. Thus, I doubt, that the indicators are continuously representative for whole of Germany. However, no information is given on spatial situation, how many stations are included in the calculation of the SPI, how are these distributed, how was the spatial aggregation undertaken? How representative for Germany is the HPI? How was spatial distributed information dealt with?

Answer: Figure 1 describes the spatial distribution of the historical information. The distribution of the given historical information covers large parts of modern Germany and neighboring regions very well.

The modern reference data are taken from the official integrated DWD data for Germany. There were no separate or additional calculations done at this stage.

3) It is additionally unclear for what the presented drought indicators can be used and what analyses can be based on the developed time series.

Answer: The presented drought reconstruction is the attempt to connect the historically derived with the modern indices and categories. Their use is the same as the modern ones, an evaluation of drought in the long-term development.

The calculation of these indicators should be complemented with analyses of the time series and their interpretation. It is quite strange, that the results and discussion section is rather a review and quite descriptive text only loosely connected with the indicators described in the methods section. I suggest to rewrite the results section completely. It should present analyses of the developed indicators and time-series and their interpretation.

Answer: We see that this was obviously not clear in the given MS. For this we followed the suggestion and we completely re-wrote this paragraph and re-organized it.

We added trend evaluations and discussed the main long-term, mid-term and yearly variations. We also added the seasonal shifts and discussed it.

4) Also the outlook section is only very loosely connected to the rest of the manuscript. I suggest to completely re-write this chapter. It should rather contain ideas of how to further analyze the developed indicators and/or how these can be analyzed in combination with other drought information. Maybe, a conclusions section would be more relevant.

Answer: We re-wrote and re-organised also the whole section.

Further comments:

Introduction Lines 20-23: These statements should be underpinned with references.

Answer: We added references

Lines 28-30: Statements could be more specific. Long-term reconstructions of what? Why are long term reconstructions necessary from comprehensive risk assessments? Century long time-series are not really necessary for

comprehensive risk assessments, rather for temporally dynamic risk assessments.

Answer: We deleted this paragraph, as risk assessment is not in the focus of this article.

Lines 41-46: the statements should be underpinned with references.

Answer: We added (and moved) the relevant references.

Lines 47-53: It would be interesting to know, which drought indices are characterizing which drought type. If this is introduced in the introduction, this could be picked up later on for the new indices presented in the study, so that it becomes clearer for what the different indicators can be used.

Answer: We refer to different indices as usual in this context. But we decided to use the SPI, because this can be derived with the historical information. We added a short comment on this.

Line 65: Please clarify to which phases you refer here. This is rather unclear.D ata It is not fully clear to me, if all information/data described in this section is used for the analyses of the manuscript, maybe some more information which data/information has been used for what might be helpful.

Answer: As now outlined in the text and the conceptual view graph, the modern period refers to 1881-2018 while the term historical is related to 1500-1996.

Lines 98-101 This paragraph fits better into the introduction.

Answer: We followed the kind advice.

Lines 117-118. The equation for the calculation of the SPI should be provided or more detail how it has been calculated and on which data specifically."Official precipitation values for Germany" is not specific. How many stations? Daily values? Spatial aggregation etc. Or was the SPI calculated already and available in the database. This becomes not fully clear here.

Answer: We used the already existing SPI values and drought categories as provided by the DWD. The spatial aggregation etc. was done by the DWD. We think more specification is not needed here, because the references are given.

Line 182-183: It remains unclear how the monthly PIs were summed up to the HPIs and how these were transformed into SPIs. Is the MDI available in the database tambora.org?

Answer: We described it more clearly ion the re-written MS. The code is also available on the Copernicus Homepage under "Assets".

Lines 188-190 it is not clear to me how the HDI was calculated on basis of the HPI. Please elaborate on this and explain better.

Answer: We added a more specific description. Together with the conceptual view graph it should be clear now.

We thank the rev.2 for the helpful comments and corrections very much!

This manuscript is a general approach to study of drougths in historical dimension, using a large database. Different indices are implemented with overlapping to instrumental data period with more complete indices availability. Historical dimension of drought is faced with a correct approach, considering it's a complex phenomena not easy to identify and evaluate in historical time, where not all information already is available for researchers. Justification of research is also well focused, with scientific and social preoccupation becase of increasing frequencies and severities of present drought events. Definition of drougths. Authors describe from a general and integrated point of view. Avoiding conceptual problems. Correct references, and historical approach, where conceptual definitions are not so easy. A complete conceptual development could take
too pages. Context of manuscript, working on historical dimension, don't justify so detailed conceptual analysis. Use of large database avoid the massive reference of sources and data previously available for this analysis. Bibliography is updated and complete. Absolutely adjusted to proposed research. Figures are well displayed and helps to understand results of manuscript. This manuscript, is a first analysis to show potential developments of historical droughts using quantitative and quantified information.

GENERAL ASPECTS + Title is too short. A subtitle could complete definition of pro-posed analysis.

Right. We have now a longer subtitle added

Table 1. Very interesting proposal. Putting in relation drought duration with drought severity seems logic and useful to study drought in historical time, where information about definition and development of indices is not so

complete and detailed as we would like. But, just a question about it. For a large natural region, as Germany or Central Europe, proposed table of criteria of classification is enough? Area under study is enoughly coherent or homogeneous to use only one system of criteria? Authors consider it would be possible application of similar method to be applied in different natural regions? Have they explored on this matter? Extension of this method to a larger spatial scale would be a good research path. Potential application of this method in other regions seems very useful. Authors could suggest any consideration about it?

This is indeed a crucial question, we discussed within the author team several times – there are different natural landscape types- true – But we decided to adjust our approach to the existing modern DWD concept, in which the whole of Germany is represented by one indicator as pointed out in table 1. It is a question of spatial scaling.

Concerning method proposed for indexing drought phenomena, manuscript show a single construction of index. Related with previous questions. All information available for Germany is reduced to one index with proposed method. Authors consider this only index is representative of drought variability for all Germany? On the other hand, it exist any wheighting process or statistical method to generate this index considering different climatic contexts? All information is considered in a similar way or level? Any clarification about it would be useful.

As pointed out in the previous paragraph, we adjusted our approach to the modern existing drought index – to find a comparable one dimensional representation. But analyzed and used a highly resoluted monthly time scale. The focus on the time means, that we could refer to the duration and the strength which is necessary to derive the chain of effects and the socio-economic dimension.
The method itself is transferable, whenever SPI makes sense for the relevant regions - the results and conclusions in this article refer to modern Germany.

Line 206. "The consequences and impacts (of drought) on the environment and society can also be reconstructued very well". This matter has increasing interest. In-tegrated approaches for natural and social dimension of hydroclimatic extremes. But authors only mention this potential in one sentence. It could be possible additional description of these potentialities, under point of view of authors? Sources, density and diversity of available information... For example, what oppinion about complementary sources, as economic (taxation records, tithes, market oscillations of prices... or

other related aspects, as records about water resources. Any consideration about this dimension of drought impacts, would be interesting to reinforce sentence of line 206.

We have now a differentiated explanation in paragraph 3.2. where we describe the historical pathways and drought categories in detail, especially the chain of effects.

Section "Outstanding single years". Lines 244-252. A more detailed description or analysis was expected. A short relation of years with drought, not chronologically ordered, with no clear explanation about severity or duration of respective drought characteristics. Please, could you explain into text what characteristics or reasons justify for every date singularity of drought recorded? Why these years are "outstanding"?. What they have in common? Any figure about characteristics of singularity: duration? Extension?, severity? any combination of magnitudes? Considering important dimension of database tambora.org, manuscript could include a more detailed analysis about extraordinary drought events? It would be an excellent opportunity to exchange knowledge of these events to other colleagues, promoting comparative analysis in different spatiotemporal scales.

We reorganized this paragraph, explained the character of single years more detailed and even added a new derived view graph for better understanding and visualization.
There is also now a table containing the list of extreme events as derived using the different indices.

SPECIFIC ASPECTS+ Lines 128-129, 152, 214, 239.

Definition and use of concept of "cascade effects"(as impacts of droughts). Term is clear, but it could be improved with a more adjusted concept? Could be possible change "cascade" by "cumulative" effects? In fact, acascade is water flowing downstream, meanwhile impacts of drought are increasing byaddition in the same place. On the other hand, use of water-related phenomena, whendrought is an important absence/shortage of water..... it seems even ironic!

This is indeed very funny, but an often cited concept, because the duration of dryness is leading to different drought types (agrarian, hydrological and socio-.economic). We also used the term "Chain of effects" as well as "pathway", which makes it somewhat clearer.

Line 166. Exclamation sign. Better final point.
Absolutely right. Done.

Lines 167-168. Unclear. Please, complete or clarify sentences.

We reworked these sentences – and made it clearer and more understandable.

Line 172. Formula doesn't appear clearly showed in text. May be by any editing prob-lem. A black dot covers partially final part of formula.

This is indeed an editing problem, which does not appear in the original MS.

Line 265. "Prominet" by "prominent"

We corrected and rewrote the whole paragraph.

---

## Referee Report (RR1)

**DESCRIPTION**

Introduction. Correct. Research motivated by increasing frequency of phenomena.

Definition of drougths. It's a complex but complet approach. Authors describe drought from a general and integrated point of view making effort for conceptual and quantitative definition. Very fruitful, as methodological milestone for future research. Correct references for historical and instrumental approach to drought events, where conceptual definitions are not so easy to define thresholds, impacts, durations.

Use of very dense database avoid the massive reference of sources and data previously available for this analysis. Consistence of results is guaranteed.

Bibliography is updated and complete. Absolutely adjusted to proposed research.

Figures are well displayed and helps to understand results of manuscript.

A first analysis, conceptual and methodological, with a good application to Germany. Example of potential developments to be applied in other historical contexts and regions.

**GENERAL ASPECTS**

After first review of December 2019, all manuscript is acceptable.

**SPECIFIC ASPECTS**

+ Line 71. "et al" by "et al."

+ Lines 71, 143, 213, 358, 369, 421, 454, 466, 472. "Brazdil by Brázdil"

+ Line 771. Figure 9. Using dates of events as symbols produce visual noise because of overlapping of labels. May be better using dots or other graphic symbols. Dates of events are already available in other formats, as lists or tables. Figure also has a temporal scale (X-axis) to locate events.

---

## Author Response (AR2)

**Suggestions for revision or reasons for rejection (will be published if the paper is accepted for final publication)**

Dear Rüdiger Glaser and Michael Kahle,
Thank you for the revision of your manuscript "Reconstructions of Droughts in Germany since 1500 - combining hermeneutic information and instrumental records in historical and modern perspectives". The new Figure 2 and the re-structuring of the text improved the clarity of your work. However, unfortunately there are still various unclear, not well explained aspects which need further improvement. The "results discussion and conclusion" section is rather weak, it contains a lot of repetition and summary of what has been shown before. The section needs to be re-written, focussed on the results of the study. A separate conclusions section is preferred.
I suggest major revision.
I hope my following comments help to further improve the manuscript:

Lines 105-108 & 140-209: It is unclear, if the derivation of the PI index and time series is part of this study, or if it is a dataset which was already available in the database tambora.org. It is described as an available dataset, but also in a quite long section in the methods chapter. It should be described either dataset, or as one step of the study, not both.

Lines 105-108 only describe the structure of the tambora data set in general. The derivation of the PI indices are derived for this paper out of the tambora codings on a monthly scale and published in the meantime as time series in Glaser & Kahle (2019).

We followed the suggestion concerning lines 140-209 and deleted two paragraphs and presented the rest in the method section.

We deleted lines 148-163 and shortened the description of the index -3. Nevertheless we think, that it is necessary to present this hermeneutic process.

Lines 109-110: Please describe in more detail, what information these "early precipitation measurements and derived drought indices" contain and how these indices look like. Additionally, it should be described how this data was used to develop the PI; or was it only used for validating the PI?

We could refer to SPIs derived from early instrumental readings by Erfurt et al. (2019) a working group close to our team of the water network Baden-Württemberg research group.

We used these Indices for cross validation only and therefore moved this paragraph into the discussion section.

Lines 117-125: This part does not fit into the data section, it seems to be more a result of a data analyses. This part might fit better towards the end of the paper, e.g. on lines 459-466 also outstanding flood events are discussed. Thus, these two different sources/indicators pointing to severe droughts could be evaluated against each other, e.g. consistency of indications checked.

We moved these lines into the result section.
For us it is not clear why we should refer these drought years to flood events???

We have examples like 1540 and 1947 in which we had both.

Line 120: typo "outstanding"
yes,

Lines 118 & 121: the 1540 event is mentioned twice, why?
True, we deleted one.

Lines 123: What does "reported as drought years" mean in respect to the indicator development?
We moved this paragraph into the result chapter (Tab.2 and the cloud Figure 9 and referred it to drought years in literature (Glaser 2013, Erfurt et al. 2018),

Line 127: "these values" – you should better describe how the values look like, e.g. averaged monthly precipitation values for x stations in Germany (or is it only one aggregated value for Germany?)
We added "Monthly precipitation measurements in mm".

Line 129-130: Are this the SPI values? In case these should be named as such here. Otherwise, it needs to be described how this data is related to the SPI.

DO-D4  are drought definitions by the DWD (2018), as citated and presented in Table 1
The relation between D0-D4 and the relevant SPIs are indicated in Table 1. The relation is based on duration and intensity defined via different thresholds.

Lines 191-193: This is double, the information has been given before.
We deleted the relevant paragraph above and kept these lines.

Figure 2: This figure is very helpful and clarifies the process. However, it should be further improved. I suggest to include for each dataset/indicator the information how it looks like, e.g. PI 7 classes from -3 to +3; HHI unclear how many classes, from -4 to +4. This information needs to be given for all indicators mentioned (preferably with information what the extreme classes mean). If it does not fit into the Figure, you should provide an additional overview table. Also include in the figure, where the weather diaries as well as the "numerical records since 1800" come into play and how.

We created a new table with the relevant information. Therefore we transformed the abbreviation list into a new Table

Lines 194-197: Describe in more detail, what information these weather diaries contain, for which years they are available (60 years between 1500 and 1800?) and most importantly how they were used for the development of the PI. Or have they been used for validation purposes only? The same is true for the "numerical records since 1800"

We added a new Table with the relevant information of observers, location and time period as well as percentage of availability.
We used these for validation purposes.

Lines 199 ff: Describe in more detail, what information these numerical records contain, for which years they are available (1800 until today?) and most importantly how they were used for the development of the PI.

We deleted this paragraph, because we used the SPIs by Erfurt et al. 2018 from 1800 onward for further validation which we described in the discussion Chapter.

Line 203: For which is the reference period good for? Average for the class 0? What doe it mean "These numerically derived indices were combined with the hermeneutically derived ones." "Combined" how and why?
Please, see the answer above - we deleted this paragraph

Chapter 3.2 Derivation of historical pathways and drought categories: it remains unclear to me how this part of the study fits into the process of index development.

We reworked the paragraph to make it clearer: we see that the historical pathways can be mapped to the modern classification schemes one by one, because the modern drought definitions also contain descriptive paragraphs.

Lines 263-264: what do you mean with "connectivity"? I guess you refer to correlation?
Yes, we refer to correlation, as
There is a high correlation between the two parameters.

Lines 265: what do you mean by "inclination"? I guess you refer to slope?
Yes, we refer to „slope" and changed it!

Figure 4: What is the scale of the SPI and of the HPI? Y-axis labels are wrong.
Yes Y-axis must be labeled as HPI, thank you. The scale of SPI and HPI ist without dimensions.

Chapter 4 Results, discussion and conclusion: large parts of this section are a repetition or summary of what has been shown before. The whole section needs to be re-written. You need to be clear about what the results of this study are. This remains rather unclear.
A separate conclusions section is preferred.

General: It is important that you use consistent terminology throughout the manuscript. This is not given and confuses the reader (e.g. scale factor – compensation factor).
yes, we homogenized it and used the term "scale factor".

We have rewritten Chapter 4 and subdivided it into the recommended results and discussion and conclusions subchapters

We thank the rev.1 and rev.2 for the encouraging and positive comments and efforts very much!

[revised manuscript text omitted]

All records in tambora.org are numerically coded, comprising spatial, temporal and content aspects. In addition to the coded events, the original text quotes are also included in the database so that the overall context and the coding can be traced for each record.

[Fig. 1: Spatial distribution of all records referring to Central Europa in *tambora.org* since 1500 (blue), precipitation, dryness and drought records (red), and impacts and consequences (green).]

The second, modern data set used for the analysis consists of the official precipitation data for Germany from 1881 onward. These values, monthly precipitation measurements in mm, recorded, averaged and provided by the DWD (Deutscher Wetter Dienst, the official German Weather Service) from the national official network stations, representing the area of modern-day Germany. This study also draws upon the official drought categories D0-D4, their classification and their definitions based on SPIs by the DWD (2019). The relation between D0-D4 and the relevant SPIs are indicated in Table 1. The relation is based on duration and intensity defined via different thresholds. based on SPIs

**3 Methods**

The conceptual design of the analysis is given in Fig.2. It illustrates the single steps and the workflow as a whole. Each individual step is described in the following subchapters in detail.

[Fig. 2: Conceptual design of the analysis, illustrating the workflow and the single steps deriving the specific indices and their relations]

Tab.1  Abbreviations of the developed indices, developed and used in the text

PI: Precipitation Index ( -3 .. + 3) (Integer)
SPI n : Standard Precipitation Index
HSPI : Historical Standard Precipitation Index
HPIn : Historical Precipitation Index ( -15 ... +15):
MDI: Modern Drought Index, DWD drought Index, 5-scale, derived from SPIs
HDI: Historical Drought Index, 5-scale drought index, derived from HPIs
HWI: Historical Wet Index, analogous to HDI for positive hygric indices
HHI: Historical Humidity Index  (-5...+5) (HDI Combines & HWI)

| Abbreviation | Description | Range (dry .. wet) | Remark |
|---|---|---|---|
| PI | Precipitation Index | -3 .. +3 | |
| SPI | Standard Precipitation Index | -x.x .. +x.x | related to normal distribution, theoretically all values possible, usually less thanthen 4 |
| HSPI | Historical Standard | -x.x .. +x.x. | depends on HPI and |

| | | | |
|---|---|---|---|
| | Precipitation Index | | calibration of slopes |
| HPI | Historical Precipitation Index | -15 .. +15 | (theoretically -36 .. +36) |
| MDI | Modern Drought Index | 4 .. 0 | according to DWD drought categories |
| HDI | Historical Drought Index | -4.0 .. 0.0 | |
| HWI | Historical Wet Index | 0.0 .. 4.0 | |
| HHI | Historical Humidity Index | -4.0 ... +4.0 | Combines HDI & HWI |

**3.1 Derivation of the monthly Precipitation Index (PI) since 1500**

The hygric indices (PI) were derived from the written evidence of the *tambora* sources via semantic profiles, a method well established in historical climatology (Glaser 1991, 1996, 2013, Glaser & Riemann 2009, Pfister 1999, Brazdil et al., 2005). Therefore, direct hygric indications as well as the descriptions of impacts and consequences are hierarchically ordered according to their intensity and assigned to the appropriate index value. A seven-scale index scheme, ranging from -3 to +3 with index 0 representing the average situation, has proven to be appropriate for the classification of historical records (Glaser 1991, 1996, Glaser & Riemann 2009, Glaser 2013).

Direct descriptions mostly refer to the absence of rain or the corresponding lack of clouds. In many cases, the duration is also indicated. The documents related to specific consequences describe effects on harvest results, the phaenological and ecological situation, but also hydrological consequences and impacts on economy, society, and their reactions. This correlates very well with the definitions of meteorological, agrarian, hydrological, groundwater and socio-economic drought in modern classifications (NDMC 2018).

Generally, extreme events are represented by a larger number of sources in historical documents. Such information also spans wider areas, especially if the records refer to droughts. Additionally, the information is more detailed and quite often severe events are compared to previous ones. Such long-term memories persist across generations.

The hierarchical class assignment and its typical indicators for the negative precipitation indices (PI) -1 to -3 are presented as follows:

Index -1 is indicated by descriptions of a beginning rainfall deficit. There are often indications of higher damages relating to the harvest of rain-sensitive products such as hay, vegetables and other garden products.

Index -2 relates to a longer duration of lack of precipitation, prolonged heat and dryness. Average crop losses for main crops are reported as well as low water levels in smaller bodies of water and reduced spring fills. Heat stress on plants, premature leaf discoloration and the death of plant parts

are observed, also dry cracks in soils, occasional forest fires and the impairment of infrastructure, for example related to shipping and water mills.

Index -3 represents extreme dryness revealing a chain of effects: After a prolonged period of dryness and heat, the agrarian consequences include severe crop losses and even harvest failures as well as emergency slaughteries due to fodder shortages. If the dryness lasts for weeks, several months or even seasons, there are integrating effects that correpond to reports like low water levels in greater lakes, ponds and larger river systems as well as the drying up of springs and wells. In addition, reports of excessive water shortage and the appearance of "hunger stones" are common. Ecological impacts include a generally visible heat stress of the vegetation, premature leaf discoloration and the withering of plants. In addition, dry cracks in soils, dust veils, dust storms and effects of wind erosion are indicated. There are diverse descriptions of a shift of the phaenological phases, e.g. early flowering, ripening and harvest, but also expression like "wine of the century" indicate dry years. The same is true for reports of forest fires and fish kills. The impairment of infrastructure, especially the termination of shipping and the failure of mills are frequently mentioned socio-economic impacts. The direct consequences for human health and well-being are also documented, e. g. through indications of heat stress and death, increased death rates, the outbreak of epidemics, diseases and hunger crisis due to a lack of food. In addition, the reports include price increases and speculations.

ADocumented authorities´ reactions range from restrictions and regulations on water access or rationing to the declaration of a state of emergency. Societal reactions like supplications, processions, pilgrimages, increasing irrational explanations and interpretations are quite common. The sources also report begging, moving around in order to seek foodseek for food, riots and protests, theft, looting, robbery and social excesses. These integrating effects allow conclusions to the preceding months, and in many cases, the exact dates of meteorological droughts are indicated by the name day of saints.

The indexing process is similar to modern classifications and definitions of drought categories. Such modern drought categoriescatgories also take into account the descriptions of impacts and societal consequences and reactions (McKee 1993, NDMC 2018).

Weather diaries with daily records exist for more than 60 years from the period 1500 to 1800, containing also on precipitation days (Lenke 1960, Klemm 1964, 1967, Glaser & Gudd 1996, Glaser 1996, Glaser 2013). These records are compared with modern precipitation data on a monthly scale, enabling a comparison of numerical rainfall data with the classified written evidenceevidences, which serves as an additional verification and validation of the index levels (PIs).

Tab.2: Selected Observers, Location and Periods with Daily Weather Entries 1500-1800

| Observer | Location | Period | Percentage of daily data |
|---|---|---|---|
|  |  |  |  |
| J. Stoeffler, J. | Tübingen | 1507-1530 | 80% |
| Johannes Indagines | Rheingau | 1517-1519 | 90% |

| | | | |
|---|---|---|---|
| NN | Dresden | 1580/1582 | 100% |
| Leonhard III Treuttwein | Fürstenfeld | 1587-1593 | 85% |
| Kilian Leib | Rebdorf | 1513-1531 | 85% |
| Hermann IV | Hessen-Kassel | 1621-1650 | 100% |
| Gottfrid Wilhelm Leibniz | Hannover | 1678 | 100% |
| Friedrich Hoffmann | Halle | 1700 | 100% |
| Camerarius | Tübingen | 1712-1715 | 85% |

For the numerical records since 1800, we used a classification scheme based on normal distribution, index „0" reflecting the monthly average with a plus/ minus 0.75 –fold standard deviation. Index „-1" ranges betwen -0.75 and -1.5 standard deviation, Index „-2" between -1.5 and -2.25 fold standard deviation and „-3" below -2.25 standard deviation. The positive indices refer to the appropriate positive ranges. The period 1951-1980 was chosen as reference period. These numerically derived indices were combined with the hermeneutically derived ones.

[revised manuscript text omitted]